# FROM SPARSE TO SOFT MIXTURES OF EXPERTS

**Joan Puigcerver**[*]
Google DeepMind

**Carlos Riquelme**[*]
Google DeepMind

**Basil Mustafa**
Google DeepMind

**Neil Houlsby**
Google DeepMind

## ABSTRACT

Sparse mixture of expert architectures (MoEs) scale model capacity without significant increases in training or inference costs. Despite their success, MoEs suffer from a number of issues: training instability, token dropping, inability to scale the number of experts, or ineffective finetuning. In this work, we propose Soft MoE, a *fully-differentiable* sparse Transformer that addresses these challenges, while maintaining the benefits of MoEs. Soft MoE performs an implicit soft assignment by passing different weighted combinations of all input tokens to each expert. As in other MoEs, experts in Soft MoE only process a subset of the (combined) tokens, enabling larger model capacity (and performance) at lower inference cost. In the context of visual recognition, Soft MoE greatly outperforms dense Transformers (ViTs) and popular MoEs (Tokens Choice and Experts Choice). Furthermore, Soft MoE scales well: Soft MoE Huge/14 with 128 experts in 16 MoE layers has over $40\times$ more parameters than ViT Huge/14, with only 2% increased inference time, and substantially better quality.

## 1 INTRODUCTION

Larger Transformers improve performance at increased computational cost. Recent studies suggest that model size and training data must be scaled together to optimally use any given training compute budget (Kaplan et al., 2020; Hoffmann et al., 2022; Zhai et al., 2022a). A promising alternative that allows to scale models in size without paying their full computational cost is sparse mixtures of experts (MoEs). Recently, a number of successful approaches have proposed ways to sparsely activate token paths across the network in language (Lepikhin et al., 2020; Fedus et al., 2022), vision (Riquelme et al., 2021), and multimodal models (Mustafa et al., 2022).

Sparse MoE Transformers involve a discrete optimization problem to decide which modules should be applied to each token. These modules are commonly referred to as *experts* and are usually MLPs. Many techniques have been devised to find good token-to-expert matches: linear programs (Lewis et al., 2021), reinforcement learning (Bengio et al., 2015), deterministic fixed rules (Roller et al., 2021), optimal transport (Liu et al., 2022), greedy top-$k$ experts per token (Shazeer et al., 2017), or greedy top-$k$ tokens per expert (Zhou et al., 2022). Often, heuristic auxiliary losses are required to balance utilization of experts and minimize unassigned tokens. These challenges can be greater in out-of-distribution settings: small inference batch sizes, novel inputs, or in transfer learning.

We introduce Soft MoE, that overcomes many of these challenges. Rather than employing a sparse and discrete router that tries to find a good *hard* assignment between tokens and experts, Soft MoEs instead perform a *soft* assignment by mixing tokens. In particular, we compute several weighted averages of all tokens—with weights depending on both tokens and experts—and then we process each weighted average by its corresponding expert.

Soft MoE L/16 outperforms ViT H/14 on upstream, few-shot and finetuning while requiring almost half the training time, and being **2× faster at inference**. Moreover, Soft MoE B/16 matches ViT H/14 on few-shot and finetuning and outperforms it on upstream metrics after a comparable amount of training. Remarkably, Soft MoE B/16 is **5.7× faster at inference** despite having 5.5× the number of parameters of ViT H/14 (see Table 1 and Figure 5 for details). Section 4 demonstrates Soft MoE's potential to extend to other tasks: we train a contrastive model text tower against the frozen vision

---

[*]Equal contribution. The order was decided by a coin toss.

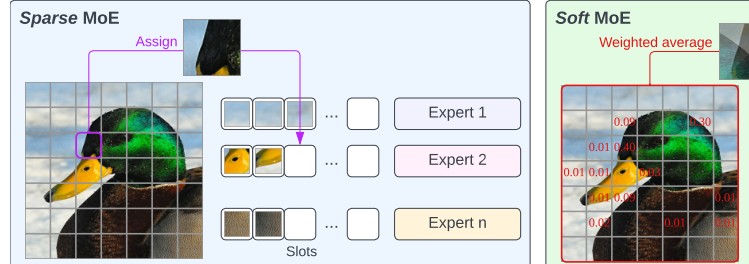

Figure 1: **Sparse and Soft MoE layers.** While the router in Sparse MoE layers (left) learns to *assign* individual input tokens to each of the available slots, in Soft MoE layers (right) each slot is the result of a (different) *weighted average* of all the input tokens. Learning to make discrete assignments introduces several optimization and implementation issues that Soft MoE sidesteps. Appendix G visualizes learned distributions of soft-assignments by Soft MoE.

tower, showing that representations learned via soft routing preserve their benefits for image-text alignment.

## 2 SOFT MIXTURE OF EXPERTS

### 2.1 ALGORITHM DESCRIPTION

The Soft MoE routing algorithm is depicted in Figure 2. We denote the inputs tokens for one sequence by $\mathbf{X} \in \mathbb{R}^{m \times d}$, where $m$ is the number of tokens and $d$ is their dimension. Each MoE layer uses a set of $n$ expert functions[1] applied on individual tokens, namely $\{f_i : \mathbb{R}^d \to \mathbb{R}^d\}_{1:n}$. Each expert processes $p$ *slots*, and each slot has a corresponding $d$-dimensional vector of parameters, $\mathbf{\Phi} \in \mathbb{R}^{d \times (n \cdot p)}$.

In particular, the input slots $\tilde{\mathbf{X}} \in \mathbb{R}^{(n \cdot p) \times d}$ are the result of convex combinations of all the $m$ input tokens, $\mathbf{X}$:

$$\mathbf{D}_{ij} = \frac{\exp((\mathbf{X}\mathbf{\Phi})_{ij})}{\sum_{i'=1}^{m} \exp((\mathbf{X}\mathbf{\Phi})_{i'j})}, \qquad \tilde{\mathbf{X}} = \mathbf{D}^\top \mathbf{X}. \tag{1}$$

Notice that $\mathbf{D}$, which we call the *dispatch* weights, is simply the result of applying a softmax over the *columns* of $\mathbf{X}\mathbf{\Phi}$. Then, as mentioned above, the corresponding expert function is applied on each slot (i.e. on rows of $\tilde{\mathbf{X}}$) to obtain the output slots: $\tilde{\mathbf{Y}}_i = f_{\lfloor i/p \rfloor}(\tilde{\mathbf{X}}_i)$.

Finally, the output tokens $\mathbf{Y}$ are computed as a convex combination of all $(n \cdot p)$ output slots, $\tilde{\mathbf{Y}}$, whose weights are computed similarly as before:

$$\mathbf{C}_{ij} = \frac{\exp((\mathbf{X}\mathbf{\Phi})_{ij})}{\sum_{j'=1}^{n \cdot p} \exp((\mathbf{X}\mathbf{\Phi})_{ij'})}, \qquad \mathbf{Y} = \mathbf{C}\tilde{\mathbf{Y}}. \tag{2}$$

We refer to $\mathbf{C}$ as the *combine* weights, and it is the result of applying a softmax over the *rows* of $\mathbf{X}\mathbf{\Phi}$.

Following the usual design for Sparse MoEs, we replace a subset of the Transformer's MLP blocks with Soft MoE blocks. We typically replace the second half of MLP blocks. The total number of slots is a key hyperparameter of Soft MoE layers because the time complexity depends on the number of slots rather than on the number of experts. One can set the number of slots equal to the input sequence length to match the FLOPs of the equivalent dense Transformer.

### 2.2 PROPERTIES OF SOFT MOE AND CONNECTIONS WITH SPARSE MOES

**Fully differentiable**  Sparse MoE algorithms involve an assignment problem between tokens and experts, which is subject to capacity and load-balancing constraints. Different algorithms approximate the solution in different ways: for example, the top-$k$ or "Token Choice" router (Shazeer et al., 2017;

---

[1]In practice, all experts apply the same function with different parameters, usually an MLP.

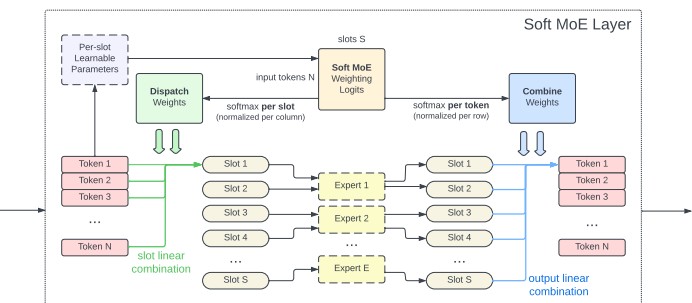

Figure 2: **Soft MoE routing details.** Soft MoE computes scores or logits for every pair of input token and slot. From this it computes a slots×tokens matrix of logits, that are normalized appropriately to compute both the dispatch and combine weights. The slots themselves are allocated to experts round-robin.

Lepikhin et al., 2020; Riquelme et al., 2021) selects the top-$k$-scored experts for each token, while there are slots available in such expert (i.e. the expert has not filled its *capacity*). The "Expert Choice" router (Zhou et al., 2022) selects the top-*capacity*-scored tokens for each expert. Other works suggest more advanced (and often costly) algorithms to compute the assignments, such as approaches based on Linear Programming algorithms (Lewis et al., 2021), Optimal Transport (Liu et al., 2022; Clark et al., 2022) or Reinforcement Learning (Clark et al., 2022). Nevertheless virtually all of these approaches are discrete in nature, and thus non-differentiable. In contrast, all operations in Soft MoE layers are continuous and fully differentiable. We can interpret the weighted averages with softmax scores as *soft* assignments, rather than the *hard* assignments used in Sparse MoE.

```python
def soft_moe_layer(X, Phi, experts):
    # Compute the dispatch and combine weights.
    logits = jnp.einsum('md,dnp->mnp', X, Phi)
    D = jax.nn.softmax(logits, axis=(0,))
    C = jax.nn.softmax(logits, axis=(1, 2))
    # The input slots are a weighted average of all the input tokens,
    # given by the dispatch weights.
    Xs = jnp.einsum('md,mnp->npd', X, D)
    # Apply the corresponding expert function to each input slot.
    Ys = jnp.stack([
        f_i(Xs[i, :, :]) for i, f_i in enumerate(experts)],
        axis=0)
    # The output tokens are a weighted average of all the output slots,
    # given by the combine weights.
    Y = jnp.einsum('npd,mnp->md', Ys, C)
    return Y
```

Algorithm 1: Simple JAX (Bradbury et al., 2018) implementation of a Soft MoE layer. Full code is available at https://github.com/google-research/vmoe.

**No token dropping and expert unbalance**    The classical routing mechanisms tend to suffer from issues such as "token dropping" (i.e. some tokens are not assigned to any expert), or "expert unbalance" (i.e. some experts receive far more tokens than others). Unfortunately, performance can be severely impacted as a consequence. For instance, the popular top-$k$ or "Token Choice" router (Shazeer et al., 2017) suffers from both, while the "Expert Choice" router (Zhou et al., 2022) only suffers from the former (see Appendix B for some experiments regarding dropping). Soft MoEs are immune to token dropping and expert unbalance since every slot is filled with a weighted average of all tokens.

**Fast**    The total number of slots determines the cost of a Soft MoE layer. Every input applies such number of MLPs. The total number of *experts* is irrelevant in this calculation: few experts with many slots per expert or many experts with few slots per expert will have matching costs if the total number of slots is identical. The only constraint we must meet is that the number of slots has to be greater or equal to the number of experts (as each expert must process at least one slot). The main advantage of Soft MoE is completely avoiding sort or top-$k$ operations which are slow and typically not well suited for hardware accelerators. As a result, Soft MoE is significantly *faster* than most sparse MoEs (Figure 6). See Section 2.3 for time complexity details.

**Features of both sparse and dense** The *sparsity* in Sparse MoEs comes from the fact that expert parameters are only applied to a subset of the input tokens. However, Soft MoEs are not technically sparse, since every slot is a weighted average of all the input tokens. Every input token *fractionally* activates all the model parameters. Likewise, all output tokens are fractionally dependent on all slots (and experts). Finally, notice also that Soft MoEs are not Dense MoEs, where every expert processes all input tokens, since every expert only processes a subset of the slots.

**Per-sequence determinism** Under capacity constraints, all Sparse MoE approaches route tokens in *groups* of a fixed size and enforce (or encourage) balance within the group. When groups contain tokens from different sequences or inputs, these tokens *compete* for available spots in expert buffers. Therefore, the model is no longer deterministic at the sequence-level, but only at the batch-level. Models using larger groups tend to provide more freedom to the routing algorithm and usually perform better, but their computational cost is also higher.

## 2.3 IMPLEMENTATION

**Time complexity** Assume the per-token cost of a single expert function is $O(k)$. The time complexity of a Soft MoE layer is then $O(mnpd + npk)$. By choosing $p = O(m/n)$ slots per expert, i.e. the number of tokens over the number of experts, the cost reduces to $O(m^2d + mk)$. Given that each expert function has its own set of parameters, increasing the number of experts $n$ and scaling $p$ accordingly, allows us to increase the total number of parameters without any impact on the time complexity. Moreover, when the cost of applying an expert is large, the $mk$ term dominates over $m^2d$, and the overall cost of a Soft MoE layer becomes comparable to that of applying a single expert on all the input tokens. Finally, even when $m^2d$ is not dominated, this is the same as the (single-headed) self-attention cost, thus it does not become a bottleneck in Transformer models. This can be seen in the bottom plot of Figure 6 where the throughput of Soft MoE barely changes when the number of experts increases from 8 to 4 096 experts, while Sparse MoEs take a significant hit.

**Normalization** In Transformers, MoE layers are typically used to replace the feedforward layer in each encoder block. Thus, when using pre-normalization as most modern Transformer architectures (Domhan, 2018; Xiong et al., 2020; Riquelme et al., 2021; Fedus et al., 2022), the inputs to the MoE layer are "layer normalized". This causes stability issues when scaling the model dimension $d$, since the softmax approaches a one-hot vector as $d \to \infty$ (see Appendix E). Thus, in Line 3 of algorithm 1 we replace X and Phi with l2_normalize(X, axis=1) and scale * l2_normalize(Phi, axis=0), respectively; where scale is a trainable scalar, and l2_normalize normalizes the corresponding axis to have unit (L2) norm, as Algorithm 2 shows.

```
1 def l2_normalize(x, axis, eps=1e-6):
2   norm = jnp.sqrt(jnp.square(x).sum(axis=axis, keepdims=True))
3   return x * jnp.reciprocal(norm + eps)
```

Algorithm 2: JAX implementation of the L2 normalization used in Soft MoE layers.

For relatively small values of $d$, the normalization has little impact on the model's quality. However, with the proposed normalization in the Soft MoE layer, we can make the model dimension bigger and/or increase the learning rate (see Appendix E).

**Distributed model** When the number of experts increases significantly, it is not possible to fit the entire model in memory on a single device, especially during training or when using MoEs on top of large model backbones. In these cases, we employ the standard techniques to distribute the model across many devices, as in (Lepikhin et al., 2020; Riquelme et al., 2021; Fedus et al., 2022) and other works training large MoE models. Distributing the model typically adds an overhead in the cost of the model, which is not captured by the time complexity analysis based on FLOPs that we derived above. In order to account for this difference, in all of our experiments we measure not only the FLOPs, but also the wall-clock time in TPUv3-chip-hours.

## 3   IMAGE CLASSIFICATION EXPERIMENTS

**Training Pareto frontiers**. In Section 3.3 we compare dense ViT models at the Small, Base, Large and Huge sizes with their dense and sparse counterparts based on both Tokens Choice and Experts Choice sparse routing. We study performance at different training budgets and show that Soft MoE dominates other models in terms of performance at a given training cost or time.

**Inference-time optimized models**. In Section 3.4, we present longer training runs ("overtraining"). Relative to ViT, Soft MoE brings large improvements in terms of inference speed for a fixed performance level (smaller models: S, B) and absolute performance (larger models: L, H).

**Model ablations**. In Sections 3.5 and 3.6 we investigate the effect of changing slot and expert counts, and perform ablations on the Soft MoE routing algorithm.

### 3.1   TRAINING AND EVALUATION DATA

We pretrain our models on JFT-4B (Zhai et al., 2022a), a proprietary dataset that contains more than 4B images, covering 29k classes. During pretraining, we evaluate the models on two metrics: upstream validation precision-at-1 on JFT-4B, and ImageNet 10-shot accuracy. The latter is computed by freezing the model weights and replacing the head with a new one that is only trained on a dataset containing 10 images per class from ImageNet-1k (Deng et al., 2009). Finally, we provide the accuracy on the validation set of ImageNet-1k after finetuning on the training set of ImageNet-1k (1.3 million images) at 384 resolution.

### 3.2   SPARSE ROUTING ALGORITHMS

*Tokens Choice*. Every token selects the top-$K$ experts with the highest routing score for the token (Shazeer et al., 2017). Increasing $K$ typically leads to better performance at increased computational cost. Batch Priority Routing (BPR) (Riquelme et al., 2021) significantly improves the model performance, especially in the case of $K = 1$ (Appendix F, Table 7). Accordingly we use Top-$K$ routing with BPR and $K \in \{1, 2\}$. We also optimize the number of experts (Appendix F, Figure 11).

*Experts Choice*. Alternatively, experts can select the top-$C$ tokens in terms of routing scores (Zhou et al., 2022). $C$ is the buffer size, and we set $E \cdot C = c \cdot T$ where $E$ is the number of experts, $T$ is the total number of tokens in the group, and $c$ is the capacity multiplier. When $c = 1$, all tokens can be processed via the union of experts. With Experts Choice routing, it is common that some tokens are simultaneously selected by several experts whereas some other tokens are not selected at all. Figure 10, Appendix B illustrates this phenomenon. We experiment with $c = 0.5, 1, 2$.

### 3.3   TRAINING PARETO-OPTIMAL MODELS

We trained ViT-{S/8, S/16, S/32, B/16, B/32, L/16, L/32, H/14} models and their sparse counterparts. We trained several variants (varying $K$, $C$ and expert number), totalling 106 models. We trained for 300k steps with batch size 4096, resolution 224, using a reciprocal square root learning rate schedule.

Figures 3a and 3b show the results for models in each class that lie on their respective training cost/performance Pareto frontiers. On both metrics, Soft MoE strongly outperforms dense and other sparse approaches for any given FLOPs or time budget. Table 9, Appendix J, lists all the models, with their parameters, performance and costs, which are all displayed in Figure 19.

### 3.4   LONG TRAINING DURATIONS

We trained a number of models for much longer durations, up to 4M steps. We trained a number of Soft MoEs on JFT, following a similar setting to Zhai et al. (2022a). We replace the last half of the blocks in ViT S/16, B/16, L/16, and H/14 with Soft MoE layers with 128 experts, using one slot per expert. We train models ranging from 1B to 54B parameters. All models were trained for 4M steps, except for H/14, which was trained for 2M steps for cost reasons.

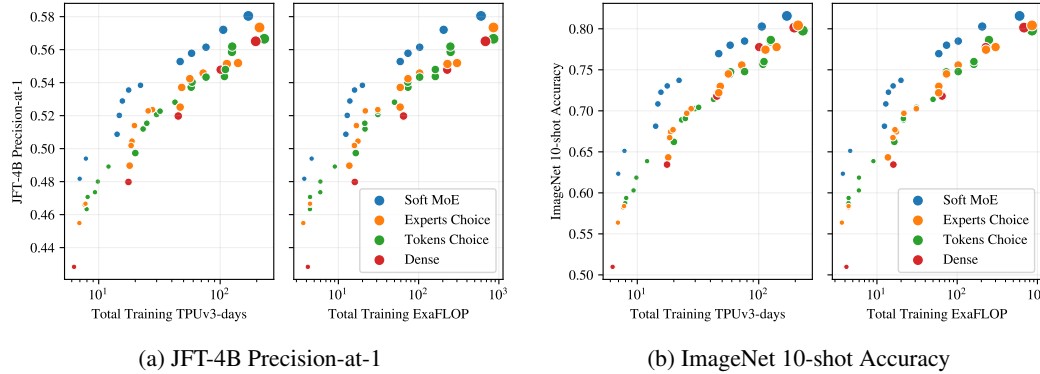

(a) JFT-4B Precision-at-1

(b) ImageNet 10-shot Accuracy

Figure 3: **Train Pareto frontiers.** Soft MoE dominates both ViTs (dense) and popular MoEs (Experts and Tokens Choice) on the training cost / performance Pareto frontier. Larger marker sizes indicate larger models, ranging from S/32 to H/14. Cost is reported in terms of FLOPS and TPU-v3 training time. Only models on their Pareto frontier are displayed, Appendix F shows all models trained.

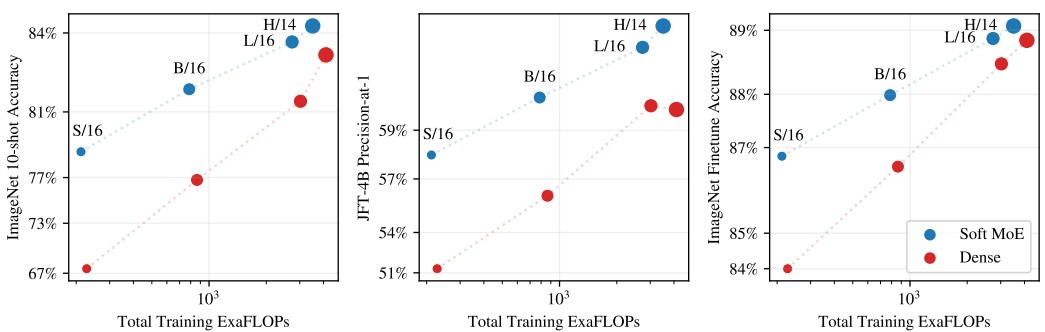

Figure 4: **Models with long training durations.** Models trained for 4M steps (H/14 trained only for 2M steps). Equivalent model classes (S/16, B/16, etc.) have similar training costs, but Soft MoE outperforms ViT on all metrics at a fixed training budget.

Figure 4 shows the JFT-4B precision, ImageNet 10-shot accuracy, and the ImageNet finetuning accuracy for Soft MoE and ViT versus training cost. Appendix F, Table 8 contains numerical results, and Figure 16 shows performance versus core-hours, from which the same conclusions can be drawn. Soft MoE substantially outperforms dense ViT models for a given compute budget. For example, the Soft MoE S/16 performs better than ViT B/16 on JFT and 10-shot ImageNet, and it also improves finetuning scores on the full ImageNet data, even though its training (and inference) cost is significantly smaller. Similarly, Soft MoE B/16 outperforms ViT L/16 upstream, and only lags 0.5 behind after finetuning while being 3x faster and requiring almost 4x fewer FLOPs. Finally, the Soft MoE L/16 model outperforms the dense H/14 one while again being around 3x faster in terms of training and inference step time.

We continue training the small backbones up to 9M steps to obtain models of high quality with low inference cost. Even after additional (over) training, the overall training time with respect to larger ViT models is similar or smaller. For these runs, longer cooldowns (linear learning rate decay) works well for Soft MoE. Therefore, we increase the cooldown from 50k steps to 500k steps.

Figure 5 and Table 1 present the results. Soft MoE B/16 trained for 1k TPUv3 days matches or outperforms ViT H/14 trained on a similar budget, and is **10× cheaper at inference** in FLOPs (32 vs. 334 GFLOPS/img) and **> 5× cheaper** in wall-clock time (1.5 vs. 8.6 ms/img). Soft MoE B/16 matches the ViT H/14 model's performance when we double ViT-H/14's training budget (to 2k TPU-days). Soft MoE L/16 **outperforms all ViT models while being almost 2× faster at inference** than ViT H/14 (4.8 vs. 8.6 ms/img).

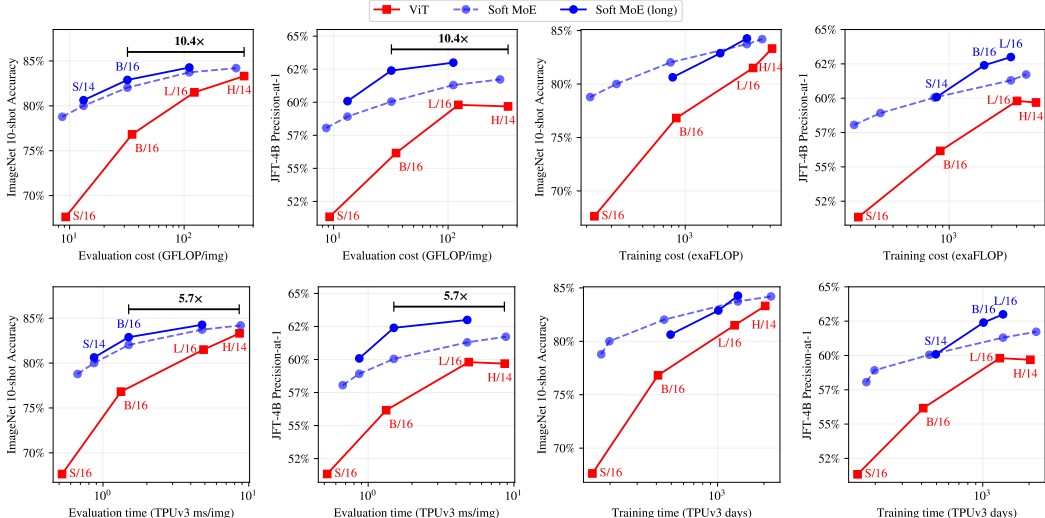

Figure 5: **Models optimized for inference speed.** Performance of models trained for more steps, thereby optimized for performance at a given inference cost (TPUv3 time or FLOPs).

Table 1: Models trained for longer durations (cooldown steps in parentheses).

| Model | Params | Train steps (cd) | Train TPU-days | Train exaFLOP | Eval ms/img | Eval GFLOP/img | JFT @1 P@1 | INet 10shot | INet finetune |
|---|---|---|---|---|---|---|---|---|---|
| ViT S/16 | 33M | 4M (50k) | 153.5 | 227.1 | 0.5 | 9.2 | 51.3 | 67.6 | 84.0 |
| ViT B/16 | 108M | 4M (50k) | 410.1 | 864.1 | 1.3 | 35.1 | 56.2 | 76.8 | 86.6 |
| ViT L/16 | 333M | 4M (50k) | 1290.1 | 3025.4 | 4.9 | 122.9 | 59.8 | 81.5 | 88.5 |
| ViT H/14 | 669M | 1M (50k) | 1019.9 | 2060.2 | 8.6 | 334.2 | 58.8 | 82.7 | 88.6 |
| ViT H/14 | 669M | 2M (50k) | 2039.8 | 4120.3 | 8.6 | 334.2 | 59.7 | 83.3 | 88.9 |
| Soft MoE S/14 256E | 1.8B | 10M (50k) | 494.7 | 814.2 | 0.9 | 13.2 | 60.1 | 80.6 | 87.5 |
| Soft MoE B/16 128E | 3.7B | 9M (500k) | 1011.4 | 1769.5 | 1.5 | 32.0 | 62.4 | 82.9 | 88.5 |
| Soft MoE L/16 128E | 13.1B | 4M (500k) | 1355.4 | 2734.1 | 4.8 | 111.1 | 63.0 | 84.3 | 89.2 |

## 3.5 NUMBER OF SLOTS AND EXPERTS

We study the effect of changing the number of slots and experts in the Sparse and Soft MoEs. Figure 6 shows the quality and speed of MoEs with different numbers of experts, and numbers of slots per token; the latter is equivalent to the average number of experts assigned per token for Sparse MoEs. When varying the number of experts, the models' backbone FLOPs remain constant, so changes in speed are due to routing costs. When varying the slots-per-expert, the number of tokens processed in the expert layers increases, so the throughput decreases. First, observe that for Soft MoE, the best performing model at each number of slots-per-token is the model with the most experts (i.e. one slot per expert). For the two Sparse MoEs, there is a point at which training difficulties outweigh the benefits of additional capacity, resulting in the a modest optimum number of experts. Second, Soft MoE's throughput is approximately constant when adding more experts. However, the Sparse MoEs' throughputs reduce dramatically from 1k experts, see discussion in Section 2.2.

## 3.6 ABLATIONS

We study the impact of the components of the Soft MoE routing layer by running the following ablations: *Identity routing*: Tokens are not mixed: the first token goes to first expert, the second token goes to second expert, etc. *Uniform Mixing*: Every slot mixes all input tokens in the same way: by averaging them, both for dispatching and combining. Expert diversity arises from different initializations of their weights. *Soft / Uniform*: We learn token mixing on input to the experts to create the slots (dispatch weights), but we average the expert outputs. This implies every input token is

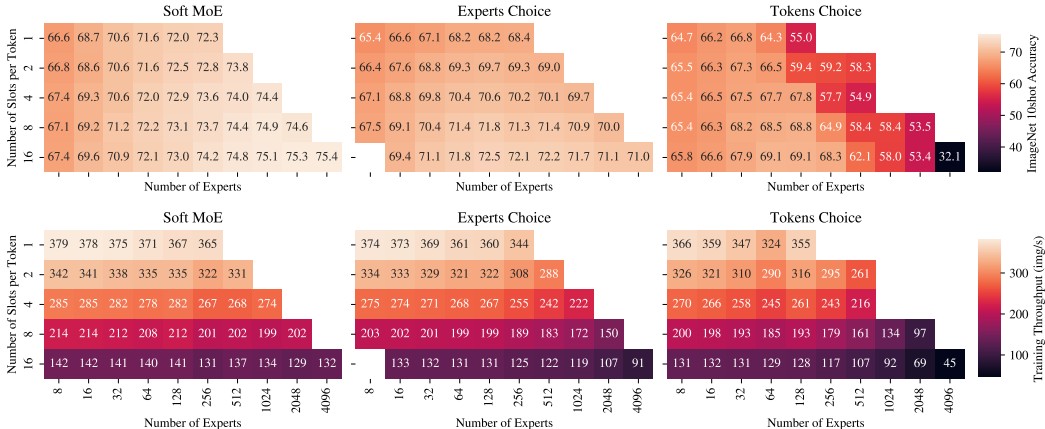

Figure 6: **Top**: Performance (ImageNet) for MoEs with different number of experts (columns) and slots-per-token / assignments-per-token (rows). **Bottom**: Training throughput of the same models. Across the columns, the number of parameters increases, however, the theoretical cost (FLOPS) for the model (not including routing cost) remains constant. Descending the rows, the expert layers become more compute intensive as more tokens/slots are processed in the MoE layers.

Table 2: Ablations using Soft MoE-S/14 with 256 experts trained for 300k steps.

| Method | Experts | Mixing | Learned Dispatch | Learned Combine | JFT p@1 | IN/10shot |
|--------|---------|--------|------------------|-----------------|---------|-----------|
| Soft MoE | ✓ | ✓ | ✓ | ✓ | 54.3% | 74.8% |
| Soft / Uniform | ✓ | ✓ | ✓ | | 53.6% | 72.0% |
| Uniform / Soft | ✓ | ✓ | | ✓ | 52.6% | 71.8% |
| Uniform | ✓ | ✓ | | | 51.8% | 70.0% |
| Identity | ✓ | | | | 51.5% | 69.1% |
| ViT | | | | | 48.3% | 62.3% |

identically updated before the residual connection. *Uniform / Soft*. All slots are filled with a uniform average of the input tokens. We learn slot mixing of the expert output tokens depending on the input tokens. Table 2 shows that having slots is important; Identity and Uniform routing substantially underperform Soft MoE, although they do outperform ViT. Dispatch mixing appears slightly more important than the combine mixing. See Appendix A for additional details.

## 4 CONTRASTIVE LEARNING

We test whether the Soft MoE's representations are better for other tasks. For this, we try image-text contrastive learning. Following Zhai et al. (2022b), the image tower is pre-trained on image classification, and then frozen while training the text encoder on a dataset of image-text pairs. We re-use the models trained on JFT in the previous section and compare their performance zero-shot on downstream datasets. For contrastive learning we train on WebLI (Chen et al., 2022), a proprietary dataset consisting of 10B images and alt-texts. The image encoder is frozen, while the text encoder is trained from scratch.

Table 3 shows the results. Overall, the benefits we observed on image classification are also in this setting. For instance, Soft MoE-L/16 outperforms ViT-L/16 by more than 1% and 2% on ImageNet and Cifar-100 zero-shot, respectively. However, the improvement on COCO retrieval are modest, and likely reflects the poor alignment between features learned on closed-vocabulary JFT and this open-vocabulary task.

Finally, in Appendix F.1 we show that Soft MoEs also surpass vanilla ViT and the Experts Choice router when trained from scratch on the publicly available LAION-400M (Schuhmann et al., 2021). With this pretraining, Soft MoEs also benefit from data augmentation, but neither ViT nor Experts Choice seem to benefit from it, which is consistent with our observation in Section 3.5, that Soft MoEs make a better use of additional expert parameters.

Table 3: LIT-style evaluation with a ViT-g text tower trained for 18B input images ($\sim$ 5 epochs).

| Model | Experts | IN/0shot | Cifar100/0shot | Pet/0shot | Coco Img2Text | Coco Text2Img |
|---|---|---|---|---|---|---|
| ViT-S/16 | – | 74.2% | 56.6% | 94.8% | 53.6% | 37.0% |
| Soft MoE-S/16 | 128 | 81.2% | 67.2% | 96.6% | 56.0% | 39.0% |
| Soft MoE-S/14 | 256 | 82.0% | 75.1% | 97.1% | 56.5% | 39.4% |
| ViT-B/16 | – | 79.6% | 71.0% | 96.4% | 58.2% | 41.5% |
| Soft MoE-B/16 | 128 | 82.5% | 74.4% | 97.6% | 58.3% | 41.6% |
| ViT-L/16 | – | 82.7% | 77.5% | 97.1% | 60.7% | 43.3% |
| Soft MoE-L/16 | 128 | 83.8% | 79.9% | 97.3% | 60.9% | 43.4% |
| Souped Soft MoE-L/16 | 128 | 84.3% | 81.3% | 97.2% | 61.1% | 44.5% |
| ViT-H/14 | – | 83.8% | 84.7% | 97.5% | 62.7% | 45.2% |
| Soft MoE-H/14 | 256 | 84.6% | 86.3% | 97.4% | 61.0% | 44.8% |

## 5 RELATED WORK

Many existing works *merge*, *mix* or *fuse* input tokens to reduce the input sequence length (Jaegle et al., 2021; Ryoo et al., 2021; Renggli et al., 2022; Wang et al., 2022), typically using attention-like weighted averages with fixed keys, to try to alleviate the quadratic cost of self-attention with respect to the sequence length. Although our dispatch and combine weights are computed in a similar fashion to these approaches, our goal is not to reduce the sequence length (while it is possible), and we actually recover the original sequence length after weighting the experts' outputs with the *combine weights*, at the end of each Soft MoE layer.

Multi-headed attention also shows some similarities with Soft MoE, beyond the use of softmax in weighted averages: the $h$ different *heads* can be interpreted as different (linear) experts. The distinction is that, if $m$ is the sequence length and each input token has dimensionality $d$, each of the $h$ heads processes $m$ vectors of size $d/h$. The $m$ resulting vectors are combined using different weights for each of the $m'$ output tokens (i.e. the attention weights), on each head independently, and then the resulting $(d/h)$-dimensional vectors from each head are concatenated into one of dimension $d$. Our experts are non-linear and combine vectors of size $d$, at the *input and output* of such experts.

Other MoE works use a weighted combination of the experts parameters, rather than doing a sparse routing of the examples (Yang et al., 2019; Tian et al., 2020; Muqeeth et al., 2023). These approaches are also fully differentiable, but they can have a higher cost, since 1) they must average the parameters of the experts, which can become a time and/or memory bottleneck when experts with many parameters are used; and 2) they cannot take advantage of vectorized operations as broadly as Soft (and Sparse) MoEs, since *every input uses a different weighted combination of the parameters*. We recommend the "computational cost" discussion in Muqeeth et al. (2023).

## 6 CURRENT LIMITATIONS

**Auto-regressive decoding**   One of the key aspects of Soft MoE consists in learning the merging of all tokens in the input. This makes the use of Soft MoEs in auto-regressive decoders difficult, since causality between past and future tokens has to be preserved during training. Although causal masks used in attention layers could be used, one must be careful to not introduce any correlation between token and slot *indices*, since this may bias which token indices each expert is trained on. The use of Soft MoE in auto-regressive decoders is a promising research avenue that we leave for future work.

**Lazy experts & memory consumption**   We show in Section 3 that one slot per expert tends to be the optimal choice. In other words, rather than feeding one expert with two slots, it is more effective to use two experts with one slot each. We hypothesize slots that use the same expert tend to align and provide small informational gains, and a expert may lack the flexibility to accommodate very different slot projections. We show this in Appendix I. Consequently, Soft MoE can leverage a large number of experts and—while its cost is still similar to the dense backbone—the memory requirements of the model can grow large.

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
