## A   SOFT VS. UNIFORM VS. IDENTITY DISPATCH AND COMBINE WEIGHTS

In this section, we compare Soft MoE (i.e. the algorithm that uses the dispatch and combine weights computed by Soft MoE in eq. (1) and eq. (2)) with different "fixed routing" alternatives, where neither the expert selected nor the weight of the convex combinations depend on the *content* of the tokens.

We consider the following simple modifications of Soft MoE:

**Identity**. The first token in the sequence is processed by the first expert, the second token by the second expert, and so on in a round robin fashion. When the sequence length is the same as the number of slots and experts, this is equivalent to replacing the matrix $\mathbf{D}$ in eq. (1) (resp. $\mathbf{C}$ in eq. (2)) with an identity matrix.

**Uniform**. Every input slot is filled with a uniform average of all input tokens, and every output token is a uniform average of all output slots. This is equivalent to replacing the matrix $\mathbf{D}$ from eq. (1) with values $\frac{1}{m}$ in all elements, and a matrix $\mathbf{C}$ from eq. (2) with values $\frac{1}{np}$ in all elements. We randomly and independently initialize every expert.

**Uniform / Soft**. Every input slot is filled with a uniform average of all input tokens, but we keep the definition of $\mathbf{C}$ from eq. (2).

**Soft / Uniform**. Every output token is a uniform average of all output slots, but we keep the definition of $\mathbf{D}$ in eq. (1).

Figure 7 and Table 2 shows the results from this experiment, training a S/14 backbone model with MoEs on the last 6 layers. Since the sequence length is 256, we choose 256 experts and slots (i.e. 1 slot per expert), so that the matrices $\mathbf{D}$ and $\mathbf{C}$ are squared. As shown in the figure, Soft MoE is far better than all the other alternatives. For context, we also add the dense ViT S/14 to the comparison.

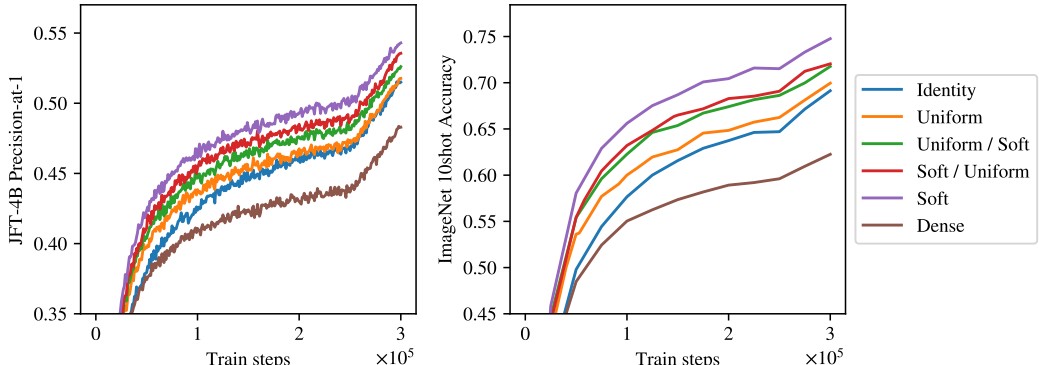

Figure 7: **Soft MoE compared against different "fixed routing" strategies**. *Identity* processes the $i$-th token with the $i$-th expert; *Uniform* replaces both the dispatch and combine matrices with uniform averages; *Uniform / Soft* replaces the dispatch weights with a uniform average, but the combine weights are computed as in Soft MoE; *Soft / Uniform* does the opposite replacement; and *Soft* uses the algorithm we present in Section 2.

## B   TOKEN DROPPING

In this appendix, we briefly explore token dropping for the Experts Choose and Tokens Choose algorithms. For Tokens Choose, each token selects $K$ experts. When experts are full, some tokens assigned to that expert will not be processed. A token is "dropped" when none of its choices go through, and no expert at all processes the token. Expert Choose algorithms lead to an uneven amount of processing per token: some input tokens are selected by many experts, while some others are not selected by any. We usually define the number of tokens to be processed by each expert in a way that the combined capacity of all experts corresponds to the number of input tokens (or a multiple $C$ of them). If we use a multiplier $C$ higher than one (say, 2x or 3x), the amount of dropping will decrease

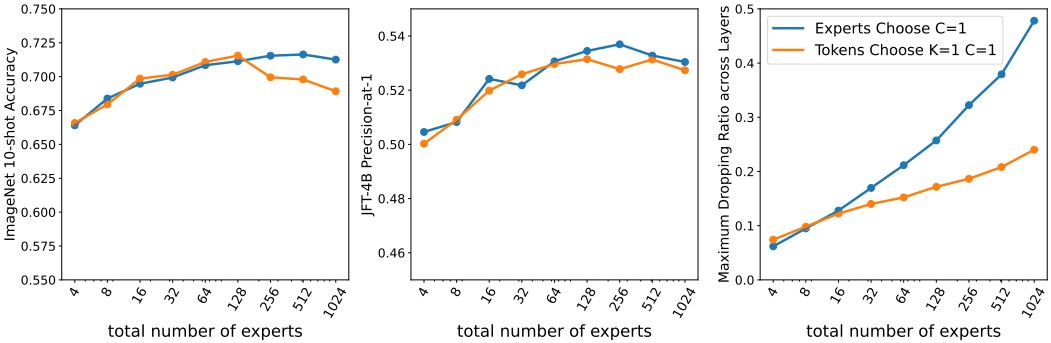

Figure 8: S/14. Performance and amount of token dropping for increasing experts for Experts Choose ($C = 1$) and Tokens Choose ($K = 1$ and $C = 1$).

but we will pay an increased computational cost. Thus, we mainly explore the $K = 1$ and $C = 1$ setup, where there is no slack in the buffers.

In all cases to follow we see a common trend: fixing everything constant, increasing the number of experts leads to more and more dropping both in Experts Choose and Tokens Choose.

Figure 8 compares Experts Choose and Tokens Choose with the same multiplier $C = 1$. This is the cheapest setup where every token *could* be assigned to an expert with balanced routing. We see that in both cases the amount of dropping quickly grows with the number of experts. Moreover, even though Experts Choose has higher levels of dropping (especially for large number of experts), it is still more performant than Tokens Choose. Note there is a fundamental difference: when Tokens Choose drops a token, the model wastes that amount of potential compute. On the other hand, for Experts Choose dropping just means some other token got that spot in the expert buffer, thus the model just transferred compute from one unlucky token to another lucky one.

In this setup, for a small number of experts (16-32) it is common to observe a $\sim 15\%$ rate of dropping. On the other hand, we also experimented with a large number of experts (100-1000) where each expert selects very few tokens. In this case, the dropping rate for Experts Choose can grow above 40-50% in some layers: most experts select the very same tokens. Tokens Choose seems to completely drop up to $\sim 25\%$ of the tokens.

In Figures 9 and 10 we study how much a little bit of buffer slack ($C = 1.125$) can help in terms of performance and dropping to Experts Choose and Tokens Choose, respectively. Both plots are similar: the amount of dropping goes down around $\sim 5\%$ and performance slightly increases when the number of experts is large. Note that the step time also increases in these cases.

Finally, Figure 11 shows the effect of Batch Priority Routing (Riquelme et al., 2021) for Tokens Choose. By smartly selecting which tokens to drop we do not only uniformly reduce the amount of dropping, but we significantly bump up performance.

## C  SOFT MOE INCREASING SLOTS

In this section we explore the following question: for a fixed number of experts, how much does Soft MoE routing benefit from having additional slots per expert? Figure 12 shows results for Soft MoE S/16 with 32 experts. We also show Experts Choice with group sizes of one and eight images. When increasing the number of slots, the performance grows only modestly, while cost increases quickly. Experts Choice benefits much more from increased slots, catching up at a large group size, but at a very large cost.

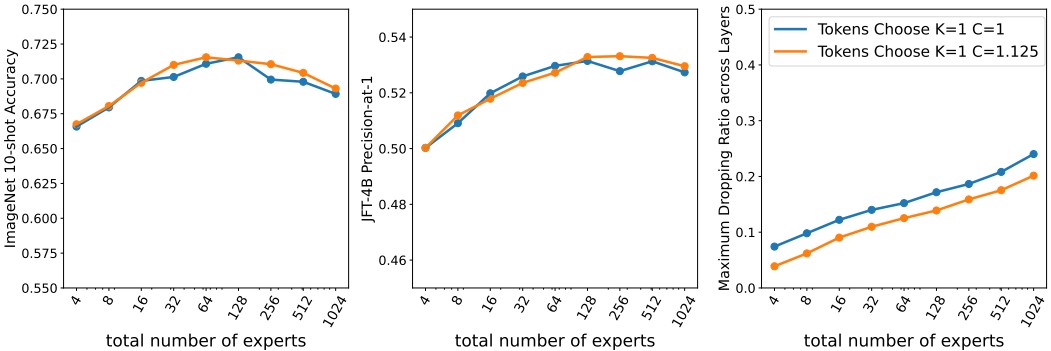

Figure 9: S/14. Performance and amount of token dropping for increasing experts for Tokens Choose with tight buffers ($K = 1$ and $C = 1$) and some amount of buffer slack ($K = 1$ and $C = 1.125$).

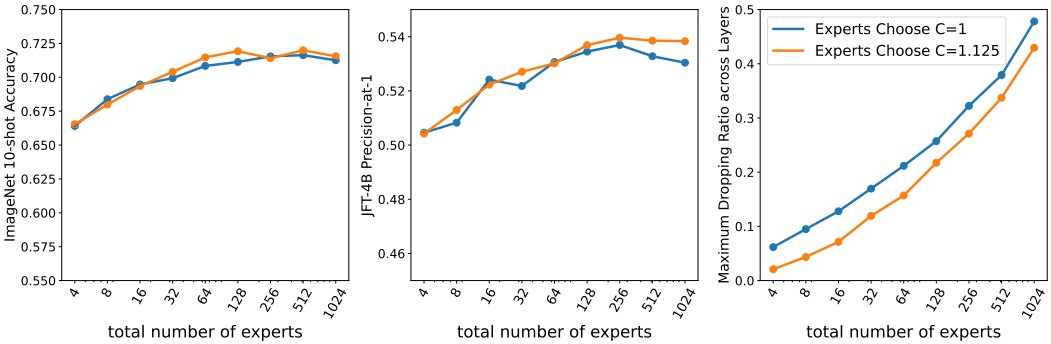

Figure 10: S/14. Performance and amount of token dropping for increasing experts for Experts Choose with tight buffers ($C = 1$) and slightly larger buffers ($C = 1.125$).

# D SPARSE LAYERS PLACEMENT

Soft MoE does unlock the effective use of a large number of experts. An important design choice for sparse models is the number and location of sparse layers, together with the number of experts per layer. Unfortunately, the large number of degrees of freedom in these choices has usually made thorough ablations and optimization unfeasible. In this section, we provide the results of a simple experiment that can help better design the configuration of sparse models. We fix a total number of experts ($E = 512$) with one slot per expert, thus leading to matched number of parameters (note in this case FLOPs may vary greatly depending on the number of sparse layers). Then, for an S/16

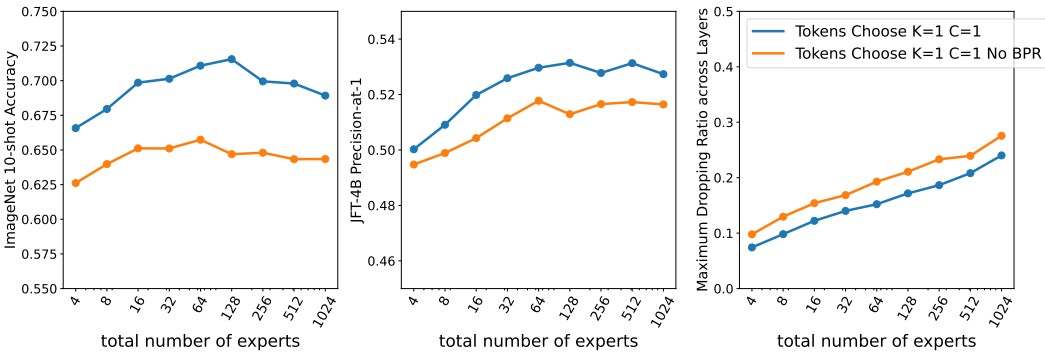

Figure 11: S/14. Performance and amount of token dropping for increasing experts with and without BPR for Tokens Choose.

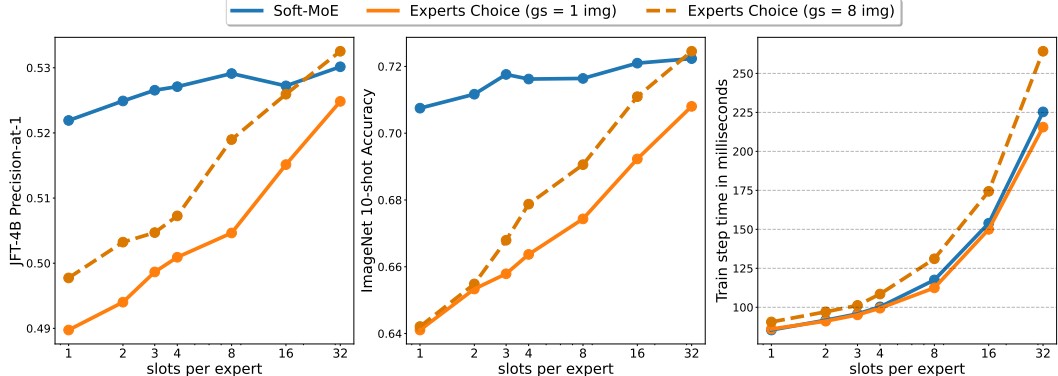

Figure 12: **Performance (left, center) and step time (right) of models with 32 experts, but increased slots, all trained for the same number of steps (300k).** Increasing the number of slots per expert only increases performance of Soft MoE a small amount, while increasing cost substantially.

backbone architecture, we distribute those experts in various ways (all in one layer, half of them in two layers, etc) and compare their performance after 300k training steps. Table 4 shows the results. Again, we observe that a number of experts close to the number of input tokens (there are 196 tokens, given the 16x16 patch size for 224x224 images) split over the last few layers works best. Moreover, note these models are indeed cheaper than those in the comparison with 512 or 256 experts per layer. Table 5 offers results for Tokens Choose routing with $K = 1$ and BPR Riquelme et al. (2021). In this case, all algorithms use a comparable FLOPs count (ignoring slightly increasing routing costs with more experts). Results are essentially similar, thus suggesting optimal expert placement (including expert count and location) may not strongly depend on the routing algorithm.

Table 4: Expert placing ablation with a Soft MoE S/16 with 12 layers (indexed from 0 to 11).

| Sparse Layers | Experts per Layer | Total Experts | IN/10shot | JFT prec1 |
| --- | --- | --- | --- | --- |
| 11 | 512 | 512 | 70.0% | 51.5% |
| 10 | 512 | 512 | 70.1% | 52.0% |
| 10, 11 | 256 | 512 | 71.7% | 52.2% |
| 5, 11 | 256 | 512 | 70.4% | 52.1% |
| 8, 9, 10, 11 | 128 | 512 | **72.8%** | **53.2%** |
| 2, 5, 8, 11 | 128 | 512 | 71.1% | 52.5% |
| 4:11 | 64 | 512 | **72.1%** | **53.1%** |
| 1:4, 8:11 | 64 | 512 | 70.5% | 52.1% |

Table 5: Expert placing ablation with a V-MoE S/16 Tokens Choose $K = 1$ with 12 layers (indexed as 0:11).

| Sparse Layers | Experts per Layer | Total Experts | IN/10shot | JFT prec1 |
|---|---|---|---|---|
| 11 | 512 | 512 | 64.4% | 50.1% |
| 10 | 512 | 512 | 67.2% | 51.9% |
| 10, 11 | 256 | 512 | 68.6% | 51.3% |
| 5, 11 | 256 | 512 | 65.3% | 50.6% |
| 8, 9, 10, 11 | 128 | 512 | **69.1%** | **52.3%** |
| 2, 5, 8, 11 | 128 | 512 | 67.3% | 51.1% |
| 4:11 | 64 | 512 | **69.9%** | **52.2%** |
| 1:4, 8:11 | 64 | 512 | 68.0% | 51.2% |

Table 6: Expert placing ablation with a V-MoE S/16 Experts Choose $C = 1$ with 12 layers (indexed as 0:11).

| Sparse Layers | Experts per Layer | Total Experts | IN/10shot | JFT prec1 |
|---|---|---|---|---|
| 11 | 512 | 512 | 65.3% | 50.3% |
| 10 | 512 | 512 | 66.5% | 51.7% |
| 10, 11 | 256 | 512 | 68.8% | 51.8% |
| 5, 11 | 256 | 512 | 65.9% | 51.1% |
| 8, 9, 10, 11 | 128 | 512 | **69.4%** | **52.2%** |
| 2, 5, 8, 11 | 128 | 512 | 68.0% | 51.7% |
| 4:11 | 64 | 512 | **69.0%** | **52.2%** |
| 1:4, 8:11 | 64 | 512 | 67.4% | 51.1% |

# E THE COLLAPSE OF SOFTMAX LAYERS APPLIED AFTER LAYER NORMALIZATION

## E.1 THEORETICAL ANALYSIS

A softmax layer with parameters $\Theta \in \mathbb{R}^{n \times d}$ transforms a vector $x \in R^d$ into the vector $\text{softmax}(\Theta x) \in \mathbb{R}^n$, with elements:

$$\text{softmax}(\Theta x)_i = \frac{\exp((\Theta x)_i)}{\sum_{j=1}^n \exp((\Theta x)_j)} = \frac{\exp(\sum_{k=1}^d \theta_{ik} x_k)}{\sum_{j=1}^n \exp(\sum_{k=1}^d \theta_{jk} x_k)} \quad (3)$$

Layer normalization applies the following operation on $x \in \mathbb{R}^d$.

$$\text{LN}(x)_i = \alpha_i \frac{x_i - \mu(x)}{\sigma(x)} + \beta_i; \quad \text{where} \ \mu(x) = \frac{1}{d} \sum_{i=1}^d x_i \ \text{and} \ \sigma(x) = \sqrt{\frac{1}{d} \sum_{i=1}^d (x_i - \mu(x_i))^2} \quad (4)$$

Notice that $\text{LN}(x) = \text{LN}(x - \mu(x))$, thus we can rewrite LayerNorm with respect to the centered vector $\tilde{x} = x - \mu(x)$, and the centered vector scaled to have unit norm $\hat{x}_i = \frac{\tilde{x}_i}{\|\tilde{x}\|}$:

$$\text{LN}(\tilde{x})_i = \alpha_i \frac{\tilde{x}_i}{\sqrt{\frac{1}{d} \sum_{j=1}^d \tilde{x}_j^2}} + \beta_i = \sqrt{d} \alpha_i \frac{\tilde{x}_i}{\|\tilde{x}\|} + \beta_i = \sqrt{d} \alpha_i \hat{x}_i + \beta_i \quad (5)$$

When a softmax layer is applied to the outputs of layer normalization, the outputs of the softmax are given by the equation:

$$\text{softmax}(\Theta\text{LN}(x))_i = \frac{\exp(\sum_{k=1}^d \theta_{ik}(\sqrt{d}\alpha_k \hat{x}_k + \beta_k))}{\sum_{j=1}^n \exp(\sum_{k=1}^d \theta_{jk}(\sqrt{d}\alpha_k \hat{x}_k + \beta_k))} \quad (6)$$

By setting $\vartheta_i = \sum_{k=1}^d \theta_{ik} \alpha_k \hat{x}_k$, and $\delta_i = \sum_{k=1}^d \theta_{ik} \beta_k$, the previous equation can be rewritten as:

$$\text{softmax}(\Theta\text{LN}(x))_i = \frac{\exp(\sqrt{d}\vartheta_i + \delta_i)}{\sum_{j=1}^n \exp(\sqrt{d}\vartheta_j + \delta_j)} \quad (7)$$

Define $m = \max_{i \in [n]} \sqrt{d}\vartheta_i - \delta_i$, $M = \{i \in [n] : \sqrt{d}\vartheta_i - \delta_i = m\}$. Then, the following equality holds:

$$\text{softmax}(\Theta\text{LN}(x))_i = \frac{\exp(\sqrt{d}\vartheta_i + \delta_i - m)}{\sum_{j=1}^n \exp(\sqrt{d}\vartheta_j + \delta_j - m)} \quad (8)$$

Given that $\lim_{d \to \infty} \exp(\sqrt{d}\vartheta_i + \delta_i - m) = \begin{cases} 1 : i \in M \\ 0 : i \notin M \end{cases}$ the output of the softmax tends to:

$$\lim_{d \to \infty} \text{softmax}(\Theta\text{LN}(x))_i = \begin{cases} \frac{1}{|M|} & i \in M \\ 0 & i \notin M \end{cases} \quad (9)$$

In particular, when the maximum is only achieved by one of the components (i.e. $|M| = 1$), the softmax collapses to a one-hot vector (a vector with all elements equal to 0 except for one).

## E.2 EMPIRICAL ANALYSIS

The previous theoretical analysis assumes that the parameters of the softmax layer are constants, or more specifically that they do not depend on $d$. One might argue that using modern parameter initialization techniques, which take into account $\frac{1}{\sqrt{d}}$ in the standard deviation of the initialization Glorot and Bengio (2010); He et al. (2015); Klambauer et al. (2017), might fix this issue. We found that they don't (in particular, we use the initialization from Glorot and Bengio (2010)).

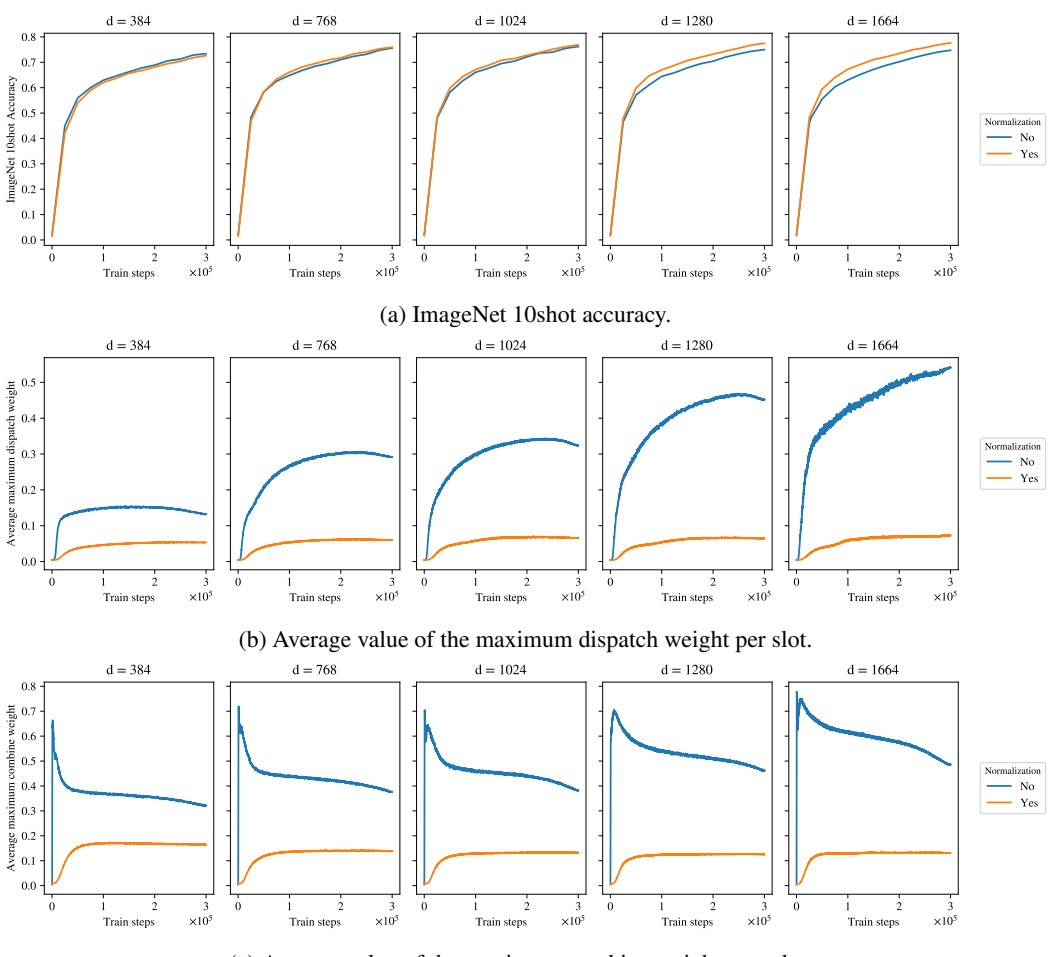

(a) ImageNet 10shot accuracy.

(b) Average value of the maximum dispatch weight per slot.

(c) Average value of the maximum combine weight per token.

Figure 13: Training plots of the ImageNet 10shot accuracy (top), the average value of the maximum dispatch weight per slot (middle) and the average value of the maximum combine weight per token (bottom) for different model dimensions $d$. Observe that maximum values of the combine and (especially) the dispatch weights grow as the model dimension grows during training, as our theoretical analysis predicted. Although the ImageNet 10shot accuracy is similar for small model dimensions, applying the softmax layer directly on the output of layer normalization, without any further re-normalization, hurts the accuracy as the model dimension $d$ grows. By normalizing the inputs to the softmax as suggested in Section 2.3 improves the performance for large values of $d$.

Figure 13 shows different metric curves during the training of a small SoftMoE model with different model dimensions. The model dimensions are those corresponding to different standard backbones: S (384), B (768), L (1024), H (1280) and G (1664). The rest of the architecture parameters are fixed: 6 layers (3 dense layers followed by 3 MoE layers with 256 experts), 14x14 patches, and a MLP dimension of 1536. As the model dimension $d$ increases, the figure shows that, if the inputs to the softmax in the SoftMoE layers are not normalized, the average maximum values of the dispatch and combine weights tend to grow (especially the former). When $d$ is big enough, the ImageNet 10shot accuracy is significantly worse than that achieved by properly normalizing the inputs.

In the previous experiment, we trained our model with a linear decay schedule and a peak value of $10^{-3}$. In addition, we also found that applying the softmax layer directly on the output of layer normalization is also very sensible to the learning rate's configuration. Once again, our recipe suggested in Section 2.3 gives equal or better quality, and is generally more stable. Figure 14 shows different metric curves during the training of the same small SoftMoE model as before, with a model dimension of $d = 1664$, using an inverse square root learning rate schedule, with a fixed timescale

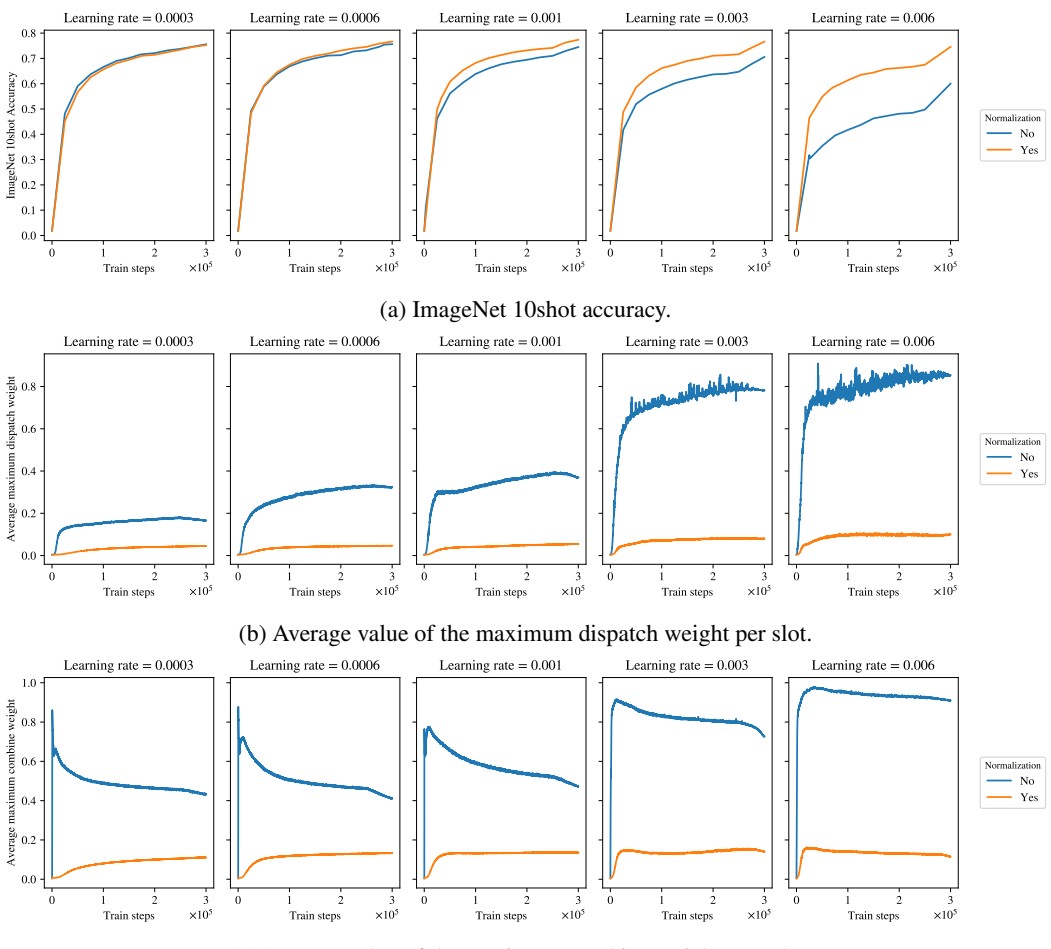

(a) ImageNet 10shot accuracy.

(b) Average value of the maximum dispatch weight per slot.

(c) Average value of the maximum combine weight per token.

Figure 14: Training plots of the ImageNet 10shot accuracy (top), the average value of the maximum dispatch weight per slot (middle) and the average value of the maximum combine weight per token (bottom) for different peak values of the learning rate, using a model dimension of $d = 1664$ (i.e. that of a G backbone).

of $10^5$, a linear warmup phase of $10^5$ steps, and a linear cooldown of $5 \cdot 10^5$ steps, varying the peak learning rate value. In this figure, similarly to the results from the previous experiment, the average maximum values of the dispatch and combine weights grows to values approaching 1.0 (indicating a collapse in the softmax layers to a one-hot vector), when the inputs to the softmax in the SoftMoE layers are not normalized, which eventually severely hurts the accuracy of the model. However, using the normalization in Section 2.3 gives better accuracy and makes the model less sensible to the choice of the peak value of the learning rate.

# F    ADDITIONAL RESULTS

## F.1    CONTRASTIVE LEARNING ON LAION-400M

We additionally train a contrastive model, similarly as described on Section 4, but training both the vision and the language towers from scratch on the publicly available dataset LAION-400M[2] (Schuhmann et al., 2021). The backbone architecture of the vision tower is a B/16. We train one model using a plain Vision Transformer, another one with MoE layers on the second half of the network with 128 experts on each layer, using an Experts Choice router, and a third model using Soft MoE. All the models use the same text tower architecture without MoEs. All the code necessary to replicate the experiments, including the training hyperparameters used, is available at https://github.com/google-research/vmoe.

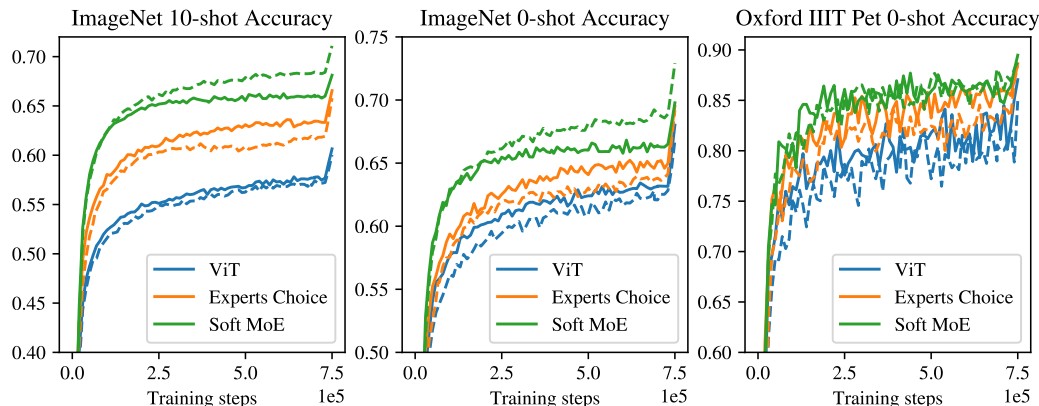

Figure 15: **Results after pretraining on LAION-400M.** Dashed lines correspond to models trained with "Inception crop" data augmentation, while solid lines correspond to models trained without any data augmentation. Soft MoE performs better than vanilla Vision Transformer (ViT) and Sparse MoE with an Experts Choice router, and benefits from data augmentation.

Figure 15 shows that, as with JFT-4B and WebLI pre-training (see Sections 3 and 4 respectively), when pretraining on the publicly available dataset LAION-400M, Soft MoE performs significantly better than both vanilla Vision Transformers and ViT with Sparse MoE layers using the Experts Choice router, across different downstream metrics. In addition, we can also see that Soft MoE benefits from data augmentation, while neither vanilla ViT or Expert Choice do. Following the observation from Figure 6 in Section 3, we hypothesize that this is because Soft MoE can better utilize the expert parameters.

## F.2    THE EFFECT OF BATCH PRIORITY ROUTING ON TOKENS CHOICE ROUTING

Table 7 shows the JFT Precision-at-1 and ImageNet 10-shot Accuracy of S/16 MoE models with Tokens Choice router, with/without using Batch Priority Routing (BPR), for different number of total experts and selected experts per token. This table shows that BPR is especially useful for $K = 1$.

## F.3    ADDITIONAL TABLES AND PLOTS COMPLEMENTING SECTION 3

---

[2]We use in fact a subset of 275M images and text pairs, since many of the original 400M examples could not be downloaded or contained corrupted images (i.e. the decoding of such images failed).

Table 7: Comparison between Top-K with and without BPR.

| Model | Number of Experts | K | BPR | JFT prec@1 | IN/10shot |
|---|---|---|---|---|---|
| V-MoE S/16 | 32 | 1 | No | 50.1% | 64.5% |
| V-MoE S/16 | 32 | 1 | Yes | 51.2% | 68.9% |
| V-MoE S/16 | 32 | 2 | No | 52.5% | 71.0% |
| V-MoE S/16 | 32 | 2 | Yes | 52.8% | 71.4% |
| V-MoE S/16 | 64 | 1 | No | 50.0% | 64.4% |
| V-MoE S/16 | 64 | 1 | Yes | 51.5% | 69.1% |
| V-MoE S/16 | 64 | 2 | No | 52.9% | 70.9% |
| V-MoE S/16 | 64 | 2 | Yes | 52.9% | 71.4% |

Table 8: Training and finetuning results for Soft MoE and dense models. Finetuning results on ImageNet at 384 resolution. We use one slot per expert and did not increase this number during finetuning, thus Soft MoEs become cheaper than ViT, as the number of input tokens grows to 576 (patch size 16x16) and 752 (patch size 14x14) but the number slots is fixed to a much smaller number (either 128 or 256).

| Model | Params | Train steps | Train days & exaFLOP | | Eval Ms/img & GFLOP/img | | JFT P@1 | IN/10s | IN/ft |
|---|---|---|---|---|---|---|---|---|---|
| ViT S/16 | 33M | 4M (50k) | 153.5 | 227.1 | 0.5 | 9.2 | 51.3 | 67.6 | 84.0 |
| Soft MoE S/16 128E | 933M | 4M (50k) | 175.1 | 211.9 | 0.7 | 8.6 | 58.1 | 78.8 | 86.8 |
| Soft MoE S/16 128E | 933M | 10M (50k) | 437.7 | 529.8 | 0.7 | 8.6 | 59.2 | 79.8 | 87.1 |
| Soft MoE S/14 256E | 1.8B | 4M (50k) | 197.9 | 325.7 | 0.9 | 13.2 | 58.9 | 80.0 | 87.2 |
| Soft MoE S/14 256E | 1.8B | 10M (500k) | 494.7 | 814.2 | 0.9 | 13.2 | 60.9 | 80.7 | 87.7 |
| ViT B/16 | 108M | 4M (50k) | 410.1 | 864.1 | 1.3 | 35.1 | 56.2 | 76.8 | 86.6 |
| Soft MoE B/16 128E | 3.7B | 4M (50k) | 449.5 | 786.4 | 1.5 | 32.0 | 60.0 | 82.0 | 88.0 |
| ViT L/16 | 333M | 4M (50k) | 1290.1 | 3025.4 | 4.9 | 122.9 | 59.8 | 81.5 | 88.5 |
| Soft MoE L/16 128E | 13.1B | 1M (50k) | 338.9 | 683.5 | 4.8 | 111.1 | 60.2 | 82.9 | 88.4 |
| Soft MoE L/16 128E | 13.1B | 2M (50k) | 677.7 | 1367.0 | 4.8 | 111.1 | 61.3 | 83.3 | 88.9 |
| Soft MoE L/16 128E | 13.1B | 4M (50k) | 1355.4 | 2734.1 | 4.8 | 111.1 | 61.3 | 83.7 | 88.9 |
| ViT H/14 | 669M | 1M (50k) | 1019.9 | 2060.2 | 8.6 | 334.2 | 58.8 | 82.7 | 88.6 |
| ViT H/14 | 669M | 2M (50k) | 2039.8 | 4120.3 | 8.6 | 334.2 | 59.7 | 83.3 | 88.9 |
| Soft MoE H/14 128E | 27.3B | 1M (50k) | 1112.7 | 1754.6 | 8.8 | 284.6 | 61.0 | 83.7 | 88.9 |
| Soft MoE H/14 128E | 27.3B | 2M (50k) | 2225.4 | 3509.2 | 8.8 | 284.6 | 61.7 | 84.2 | 89.1 |
| Soft MoE H/14 256E | 54.1B | 1M (50k) | 1276.9 | 2110.1 | 10.9 | 342.4 | 60.8 | 83.6 | 88.9 |
| Soft MoE H/14 256E | 54.1B | 2M (50k) | 2553.7 | 4220.3 | 10.9 | 342.4 | 62.1 | 84.3 | 89.1 |

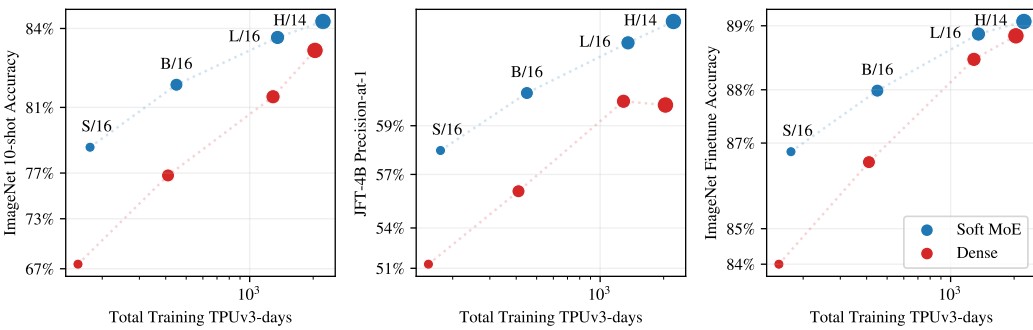

Figure 16: **Long runs.** Soft MoE and ViT models trained for 4 million steps with batch size 4096 (H/14 models trained for 2 million steps instead). Equivalent model classes (S/16, B/16, L/16, H/14) have similar training costs, but Soft MoE outperforms ViT on all metrics. We show ImageNet 10-shot (left), JFT precision at 1 (middle) and ImageNet accuracy after finetuning (right), versus total training FLOPs. See Table 8. We report training FLOPs in Figure 4.

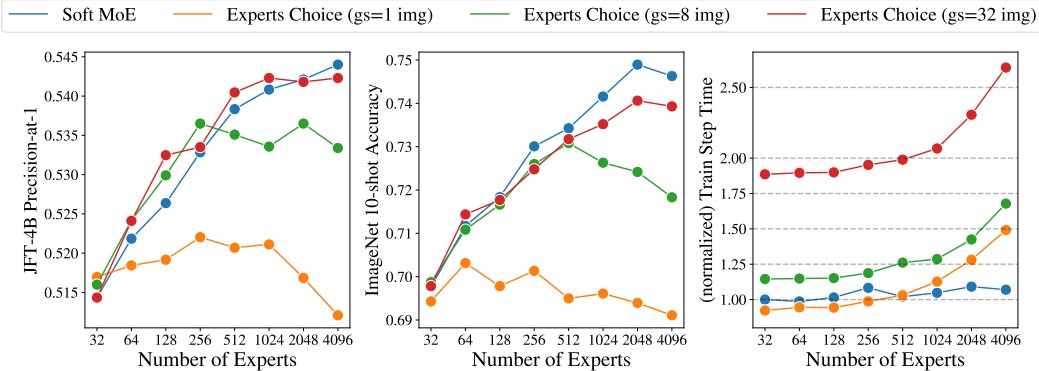

Figure 17: JFT precision-at-1, ImageNet 10-shot accuracy, and normalized training step time when increasing the total number of experts while keeping the total amount of slots fixed. Soft MoE achieves consistently better results with more experts, whereas cost is kept roughly constant. Adding too many experts to Experts Choice hurt performance and significantly increases the cost. Experts Choice can perform well with many experts if we increase the group size up to 32 images per group. The normalized train step time is computed with respect to Soft MoE with 32 experts. Experts Choice with 32 images per group and 4096 experts requires more than 2.5x its cost.

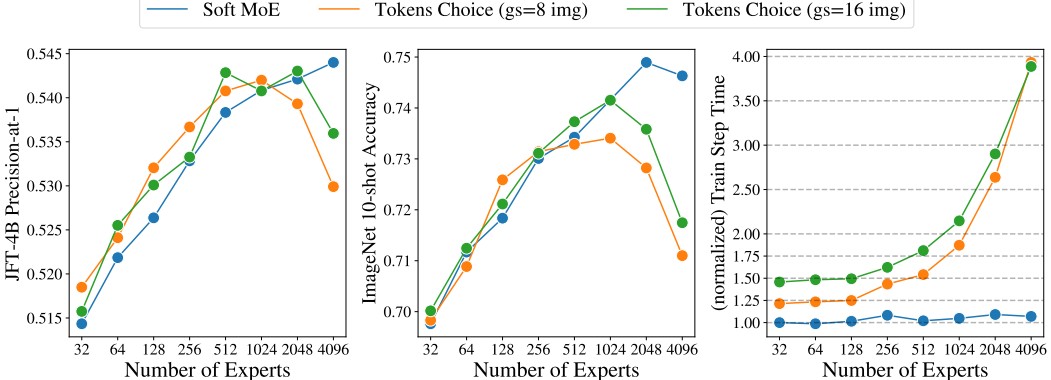

Figure 18: JFT precision-at-1, ImageNet 10-shot accuracy, and normalized training step time when increasing the total number of experts while keeping the total amount of slots fixed. Soft MoE achieves consistently better results with more experts, whereas cost is kept roughly constant. Adding too many experts to Tokens Choice hurt performance and significantly increases the cost. Even with a large group size (16 images), Tokens Choice struggles to perform well with a few thousand experts. The normalized train step time is computed with respect to Soft MoE with 32 experts. Tokens Choice with 8 or 16 images per group and 4096 experts requires almost 4x its cost.

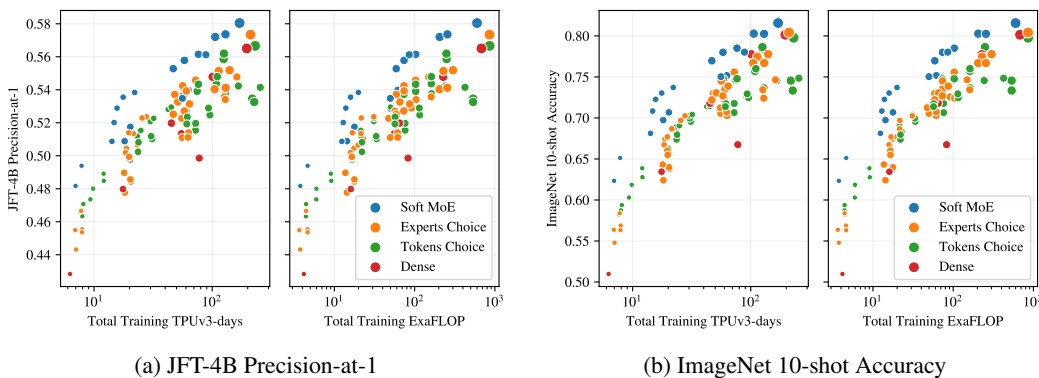

(a) JFT-4B Precision-at-1

(b) ImageNet 10-shot Accuracy

Figure 19: JFT-4B Precision-at-1 and ImageNet 10-shot accuracy on short runs (300k steps). The size of the marker depends on the backbone size: S/32, S/16, B/32, B/16, L/16 and H/14. Colors represent different methods: Soft MoE (blue), Sparse MoEs with Experts Choice (orange) and Tokens Choice routing (green), and a Dense (red) model. MoE runs include different configurations.

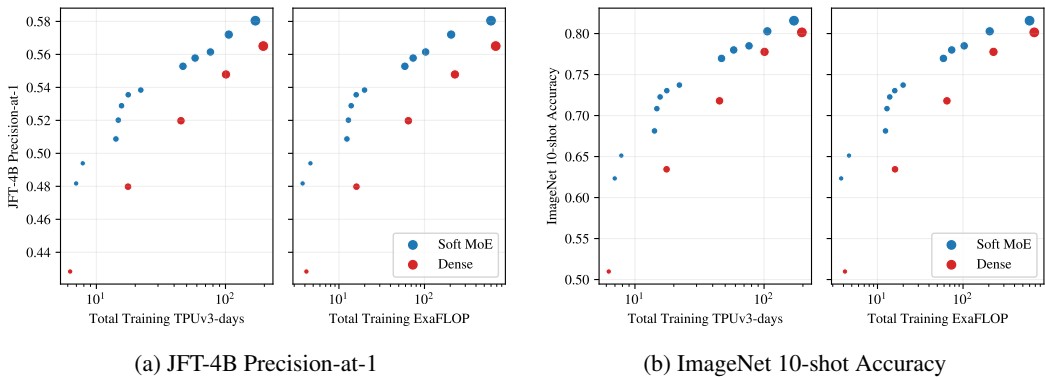

(a) JFT-4B Precision-at-1

(b) ImageNet 10-shot Accuracy

Figure 20: JFT-4B Precision-at-1 and ImageNet 10-shot accuracy on short runs (300k training steps). The size of the marker depends on the backbone size: S/32, S/16, B/32, B/16, L/16 and H/14. Colors represent different methods: Soft MoE (blue) and Dense (red) models. MoE runs include different configurations. We only show the runs that are not dominated by another model using the same method (S/8 and L/32 were always dominated).

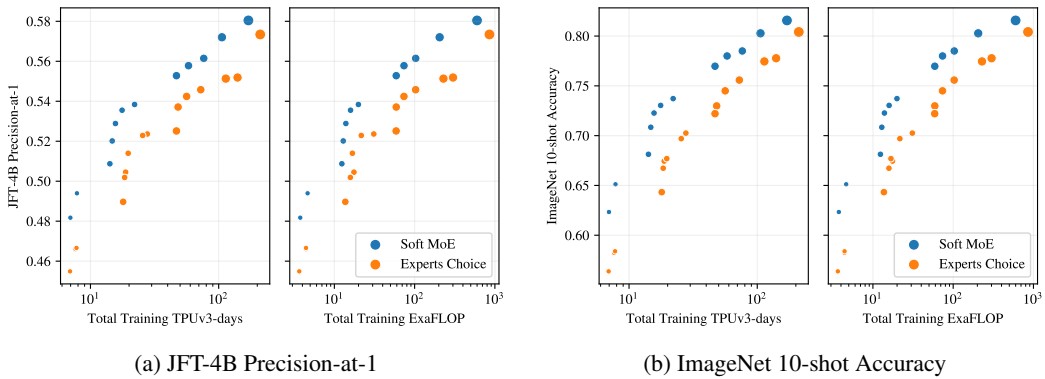

(a) JFT-4B Precision-at-1

(b) ImageNet 10-shot Accuracy

Figure 21: JFT-4B Precision-at-1 and ImageNet 10-shot accuracy on short runs (300k training steps). The size of the marker depends on the backbone size: S/32, S/16, B/32, B/16, L/16 and H/14. Colors represent different methods: Soft MoE (blue) and Sparse MoEs with Experts Choice (orange) models. MoE runs include different configurations. We only show the runs that are not dominated by another model using the same method (S/8 and L/32 were always dominated).

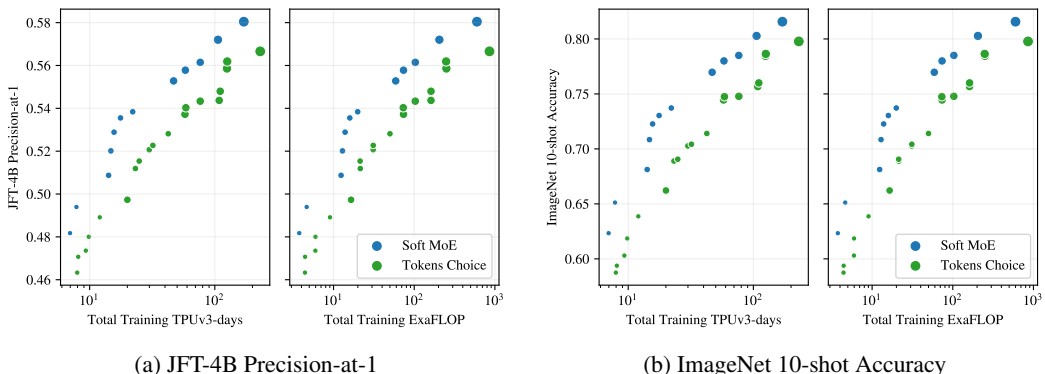

(a) JFT-4B Precision-at-1           (b) ImageNet 10-shot Accuracy

Figure 22: JFT-4B Precision-at-1 and ImageNet 10-shot accuracy on short runs (300k training steps). The size of the marker depends on the backbone size: S/32, S/16, B/32, B/16, L/16 and H/14. Colors represent different methods: Soft MoE (blue) and Sparse MoEs with Tokens Choice (green) models. MoE runs include different configurations. We only show the runs that are not dominated by another model using the same method (S/8 and L/32 were always dominated).

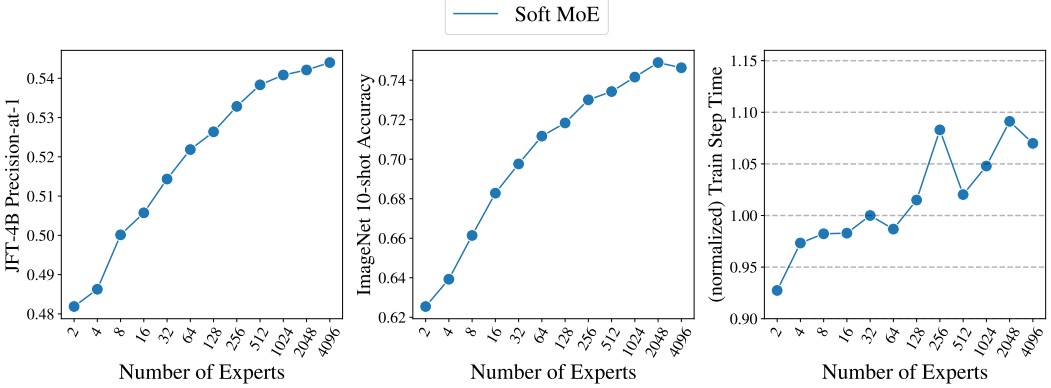

Figure 23: **JFT Precision-at-1, ImageNet 10-shot Accuracy, and normalized Training Step time when increasing the total number of experts while keeping the total amount of slots fixed (4096)**. Soft MoE achieves consistently better results with more experts, whereas cost is kept roughly constant (same FLOPs but communication costs vary due to higher topologies needed for larger models). The normalized train step time is computed with respect to Soft MoE with 32 experts. Model sizes range from 38M (2 experts) to 9.7B parameters (4096 experts).

# G  MODEL INSPECTION

In this section, we take a look at various aspects of the routing the model learns.

**Tokens contributions to slots.** While there is no dropping in Soft MoE, it is still possible that some tokens contribute little to *all* slots if their logits are much lower than those of other tokens. We would like to see if some tokens contribute to slots in a disproportionate manner. Figure 24 (left) shows the distribution across tokens for the total weight each token provides to slots (i.e. summed over all slots). This was computed over a batch with 256 images with 196 tokens each on a Soft MoE S/16 finetuned on ImageNet. We see there is a heavy tail of tokens that provide a stronger total contribution to slots, and the shape is somewhat similar across layers. Around 2-5% of the tokens provide a summed weight above 2. Also, between 15% and 20% of the tokens only contribute up to 0.25 in total weight. The last layer is slightly different, where token contribution is softer tailed. Appendix H further explores this.

**Experts contributions to outputs.** Similarly, we would like to understand how much different slots end up contributing to the output tokens. We focus on the case of one slot per expert. We can approximate the total contribution of each expert (equivalently, slot) by averaging their corresponding coefficients in the linear combinations for all output tokens in a batch. Figure 24 (center) shows such (normalized) importance across experts for different MoE layers. We see that, depending on the layer, some experts can impact output tokens between 3x and 14x more than others.

**Number of input tokens per slot.** For each slot, Figure 24 (right) shows how many input tokens are required to achieve a certain cumulative weight in its linear combination. The distribution varies significantly across slots. For a few slots the top 20-25 tokens account for 90% of the slot weight, while for other slots the distribution is more uniform and many tokens contribute to fill in the slot. In general, we see that slots tend to mix a large number of tokens unlike in standard Sparse MoEs.

**Visual inspection.** In order to provide some intuition regarding how slots average input tokens, Figure 25 graphically shows the linear combinations for 8 different slots for the image shown in Figure 1. We shade patches inversely proportionally to their weight in the slots; note that all tokens representations are eventually combined into a single one (with hidden dimension $h$) before being passed to the expert (unlike in our plot, where they are arranged in the usual way). These plots correspond to a Soft MoE S/16 with 128 experts and one slot per expert, and we handpicked 8 out of the 128 slots to highlight how different slots tend to focus on different elements of the image.

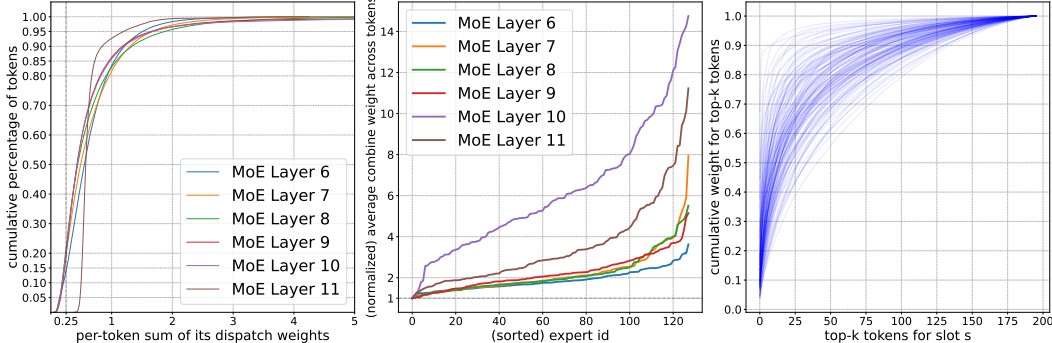

Figure 24: **(Left)** Distribution of summed dispatch weights per token for different MoE layers. For instance, in layer 11, the dispatch weights for 90-95% of the input tokens summed over all the slots are at most 1. Only a tiny fraction of tokens contribute to slots by summing more than 3. **(Middle)** Distribution of combine weights per slot (or expert, as we use one slot per expert) summed across all input tokens. We normalize the sum by its minimum value across experts. **(Right)** Each curve corresponds to one slot. Dispatch weights from all tokens to each slot add up to 1. Distribution of how many inputs tokens are needed to achieve a certain fraction of the total weight for the slot.

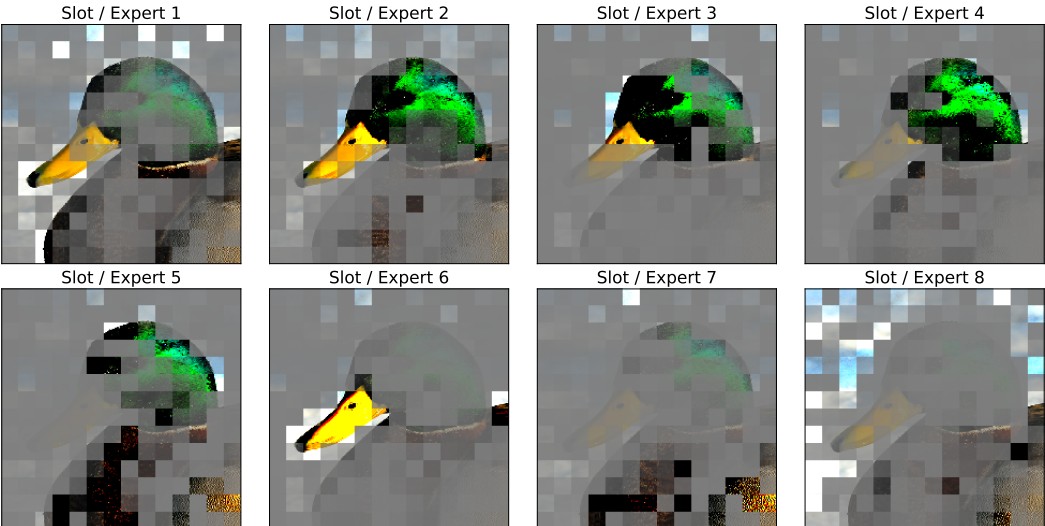

Figure 25: Linear combinations for 8 slots when using input image in Figure 1. Model is Soft MoE S/16 with 128 experts and one slot per expert, and it was finetuned on ImageNet. We show results for the first MoE layer (seventh block). The selected slots (among 128) are cherry-picked to highlight differences across slots.

## H    ADDITIONAL ANALYSIS

### H.1    CUMULATIVE SUM OF DISPATCH AND COMBINE WEIGHTS

Figure 26 shows the distribution over slots of the cumulative sum (over tokens) of their corresponding dispatch weights. For each slot we compute the cumulative sum of the dispatch weights over tokens sorted in decreasing order. This indicates how many tokens are necessary to cover a given percentage of the total mass of the weighted average. We compute this cumulative sum for all slots over all the 50 000 ImageNet validation images, across all layers of the Soft MoE H/16 model after finetuning. In the plot, we represent with a solid line the average (over all slots and images) cumulative sum, and the different colored areas represent the central 60%, 80%, 90%, 95% and 99% of the distribution (from darker to lighter colors) of cumulative sums.

This tells us, for instance, how uniform is the weighted average over tokens used to compute each input slot. In particular, each slot in the last two layers is close to a uniform average of all the tokens (a completely uniform average would be represented by a straight line). This tells us that in these layers, every expert processes roughly the same inputs, at least after the model is trained. However, this weighted average is far from uniform in the rest of the layers, meaning that there are tokens that contribute far more than others. For example, in layer 28, a few tens of tokens already cover 80% of the weighted average mass. Finally, given the width of the colored areas, we can also see that there's a significant difference on the weighted averages depending on the slot, across all layers (except maybe the last two). This indicates that the dispatch weights vary across different slots and images.

Similarly, Figure 27 shows the corresponding plots for the cumulative sum of the combine weights. In this case, for each output token we compute the cumulative sum of the combine weights over slots sorted in decreasing order. Notice that, although the dispatch weights in the last two layers were almost uniform, the combine weights are not. This indicates that some slots (and thus, experts) are more important than others in computing the output tokens, and thus their corresponding expert parameters are not redundant. Of course, the identity of the "important" slots may vary depending on the input token.

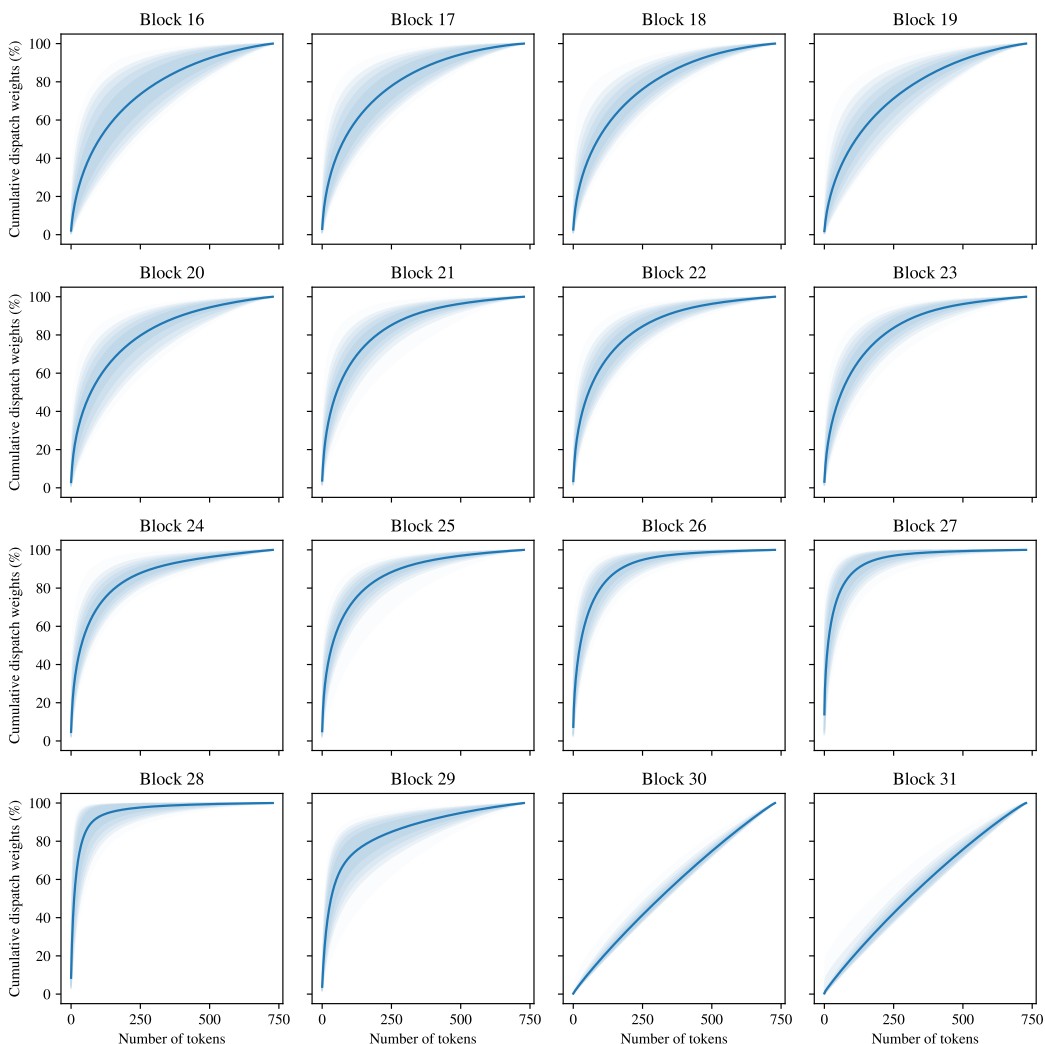

Figure 26: **Distribution of the cumulative sum of dispatch weights.** For each input slot, we compute the cumulative sum of its corresponding dispatch weights (sorted by decreasing value). This indicates over how many input tokens a certain cumulative weight is distributed over. The line in each plot represents the average computed over all slots and ImageNet validation images of the given block in the SoftMoE H/14 model. The colored areas represent the central 60%, 80%, 90%, 95% and 99% of the distribution (from darker to lighter, better seen in color).

## I  SLOT CORRELATION

In this section we explore the correlation between the different slot *parameters* that Soft MoE learns, and its relationship with the number of slots per expert. Figures 28 to 30 show for each of 6 layers in a Soft MoE S/16 the inner product between each pair of (normalized) slot parameter vectors.

While Figure 28 shows no clear relationship between slots from different experts (as each expert only has one slot), we observe in Figures 29 and 30 how consecutive slots (corresponding to the same expert) are extremely aligned. This confirms our hypothesis that adding more slots to experts does not work very well as these slots end up aligning their value, and computing somewhat similar linear combinations. Therefore, these projections do not add too much useful information to the different tokens to be processed by the experts (in the extreme, these slots would be identical).

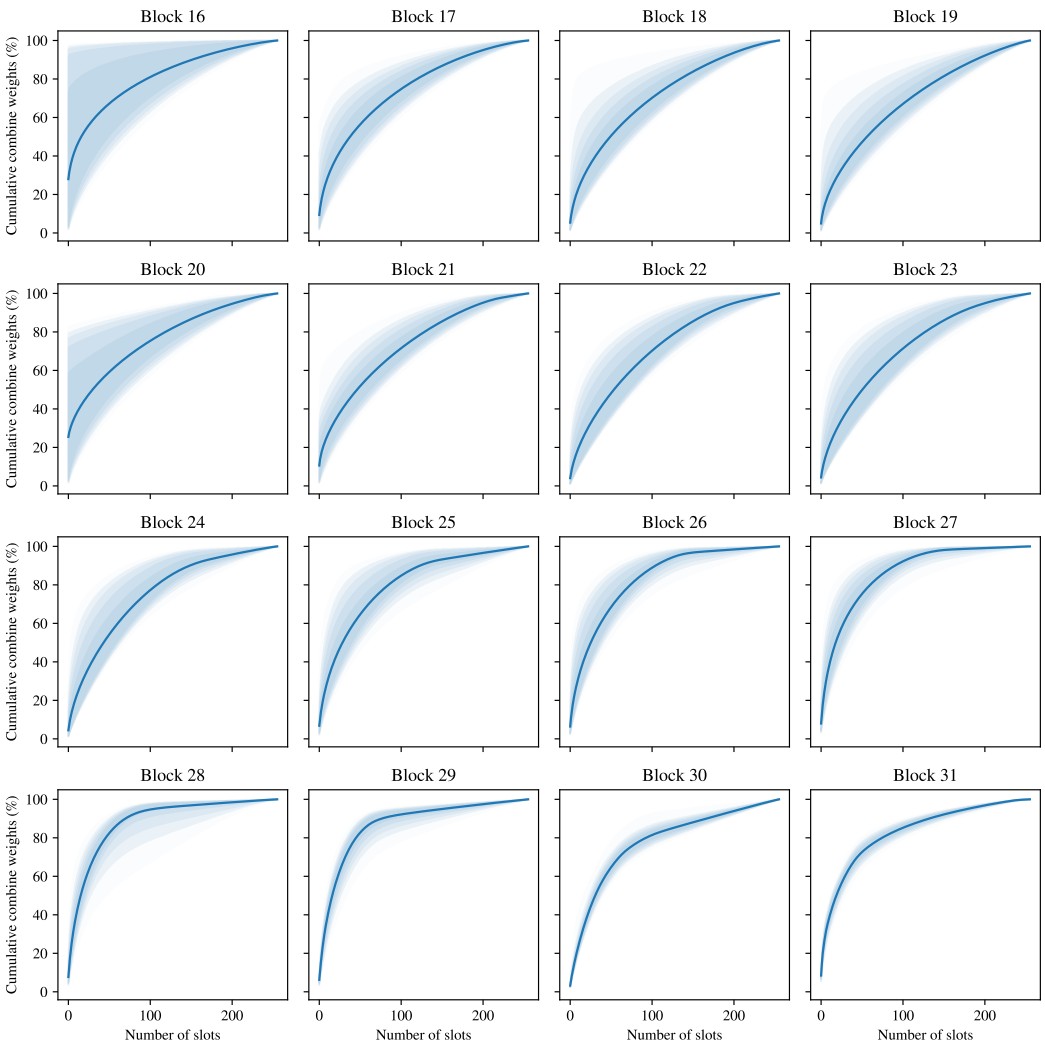

Figure 27: **Distribution of the cumulative sum of combine weights.** For each output token, we compute the cumulative sum of its corresponding combine weights (sorted by decreasing value). This indicates over how many output slots a certain cumulative weight is distributed over. The line in each plot represents the average computed over all tokens and ImageNet validation images of the given block in the SoftMoE H/14 model. The colored areas represent the central 60%, 80%, 90%, 95% and 99% of the distribution (from darker to lighter, better seen in color).

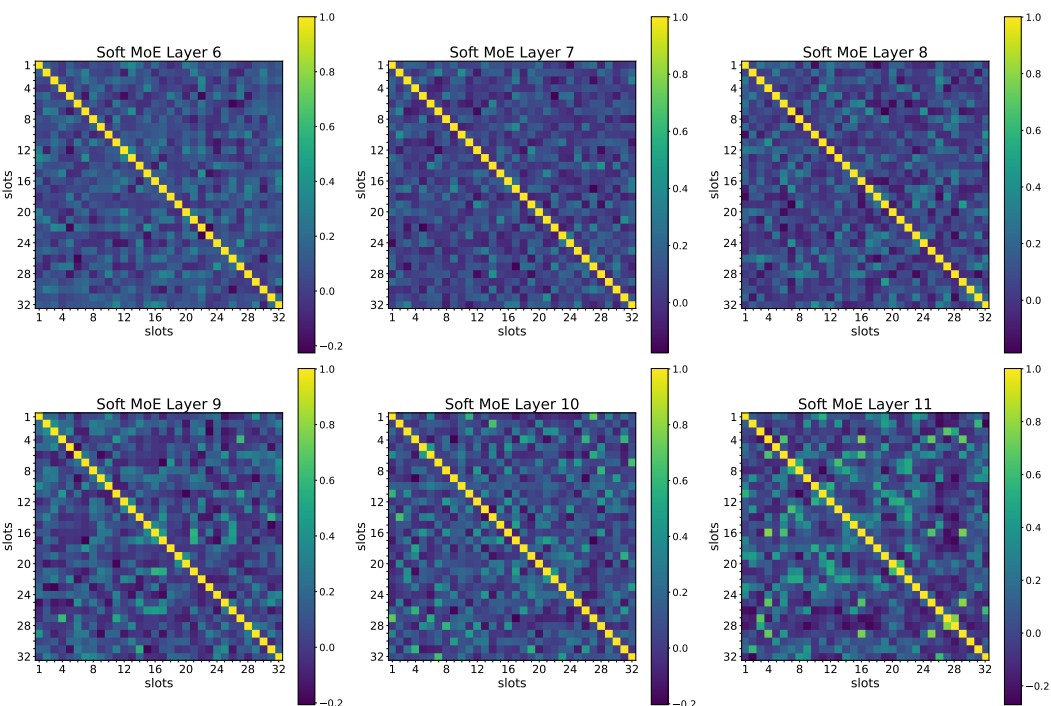

Figure 28: Soft MoE S/16 with 1 slot per expert.

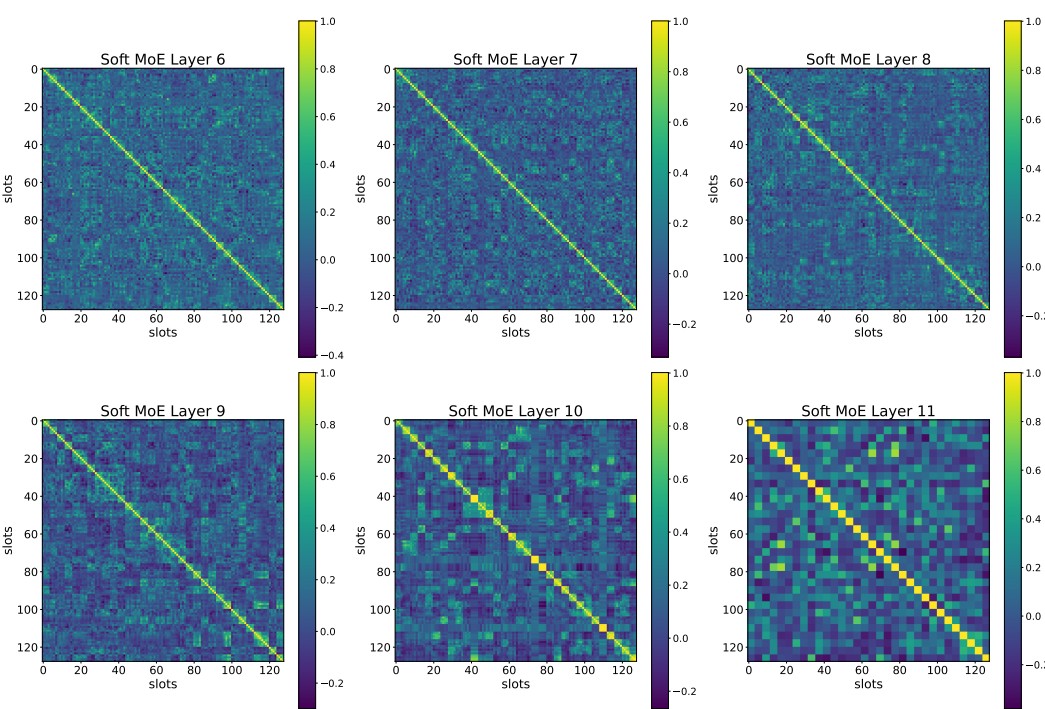

Figure 29: Soft MoE S/16 with 4 slots per expert.

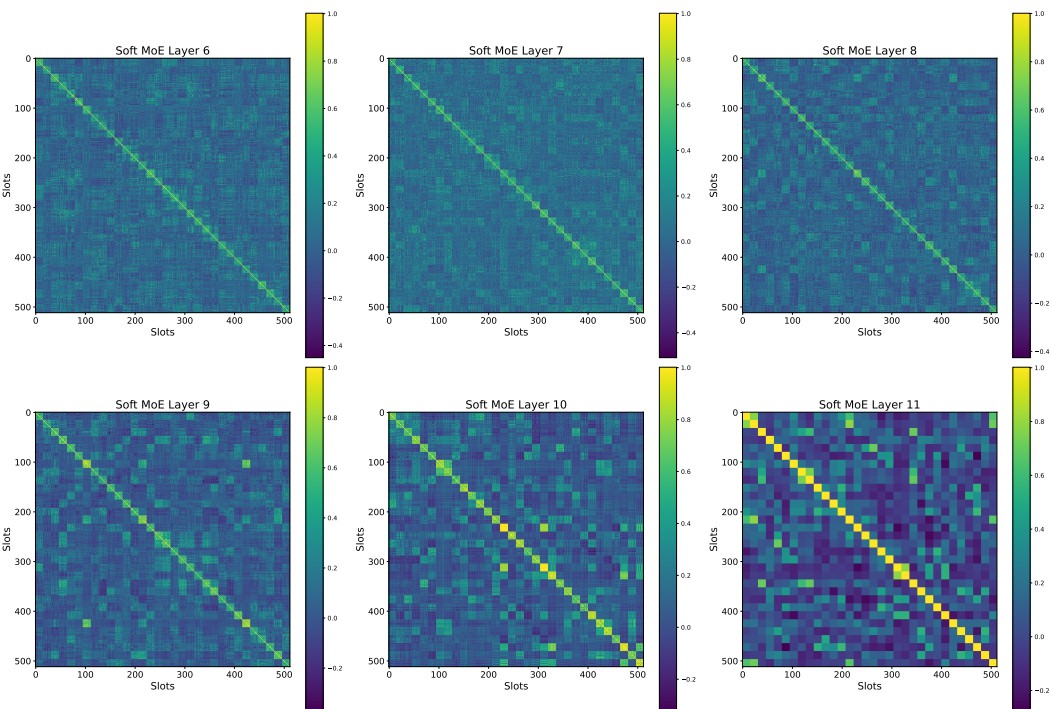

Figure 30: Soft MoE S/16 with 16 slots per expert.

## J  PARETO MODELS

Table 9: Model runs from Section 3.3 (shown in Pareto plot) trained for 300k steps on JFT with inverse square root decay and 50k steps cooldown. We trained dense and MoE (Soft MoE, Tokens Choice, Experts Choice) models with sizes S/32, S/16, S/8, B/32, B/16, L/32, L/16 and H/14. Sorted by increasing training TPUv3 days.

| Ref | Model | Routing | Experts | Group Size | K | C | JFT P@1 | IN/10shot | Train exaFLOP | Train Days |
|---|---|---|---|---|---|---|---|---|---|---|
| 1 | S/32 | Dense | – | – | – | – | 42.8 | 51.0 | 4.2 | 6.3 |
| 2 | S/32 | Experts Choice | 32 | 392 | – | 0.5 | 45.5 | 56.4 | 3.7 | 6.9 |
| 3 | S/32 | Soft MoE | 32 | 49 | – | – | 48.2 | 62.3 | 3.8 | 7.0 |
| 4 | S/32 | Experts Choice | 32 | 49 | – | 0.5 | 44.3 | 54.8 | 3.8 | 7.0 |
| 5 | S/32 | Experts Choice | 32 | 392 | – | 1.0 | 46.6 | 58.2 | 4.4 | 7.6 |
| 6 | S/32 | Experts Choice | 64 | 392 | – | 1.0 | 46.7 | 58.4 | 4.4 | 7.8 |
| 7 | S/32 | Soft MoE | 64 | 49 | – | – | 49.4 | 65.1 | 4.6 | 7.8 |
| 8 | S/32 | Experts Choice | 64 | 49 | – | 1.0 | 45.4 | 56.3 | 4.6 | 7.9 |
| 9 | S/32 | Experts Choice | 32 | 49 | – | 1.0 | 45.5 | 56.9 | 4.6 | 7.9 |
| 10 | S/32 | Tokens Choice | 32 | 392 | 1 | 1.0 | 46.3 | 58.7 | 4.4 | 8.0 |
| 11 | S/32 | Tokens Choice | 64 | 392 | 1 | 1.0 | 47.1 | 59.4 | 4.4 | 8.1 |
| 12 | S/32 | Tokens Choice | 32 | 392 | 2 | 1.0 | 47.4 | 60.3 | 6.0 | 9.3 |
| 13 | S/32 | Tokens Choice | 64 | 392 | 2 | 1.0 | 48.0 | 61.9 | 6.0 | 9.8 |
| 14 | B/32 | Dense | – | – | – | – | 48.0 | 63.4 | 16.0 | 11.7 |
| 15 | B/32 | Soft MoE | 32 | 49 | – | – | 50.9 | 69.7 | 14.3 | 11.7 |
| 16 | S/32 | Tokens Choice | 64 | 392 | 2 | 2.0 | 48.9 | 63.9 | 9.0 | 12.0 |
| 17 | S/32 | Tokens Choice | 32 | 392 | 2 | 2.0 | 48.5 | 62.8 | 9.1 | 12.1 |
| 18 | S/16 | Soft MoE | 16 | 196 | – | – | 50.9 | 68.1 | 12.4 | 14.2 |
| 19 | S/16 | Soft MoE | 32 | 196 | – | – | 52.0 | 70.8 | 12.9 | 14.8 |
| 20 | S/16 | Dense | – | – | – | – | 47.9 | 60.8 | 17.0 | 15.3 |
| 21 | S/16 | Soft MoE | 64 | 196 | – | – | 52.9 | 72.3 | 13.9 | 15.7 |
| 22 | S/16 | Soft MoE | 128 | 196 | – | – | 53.6 | 73.0 | 15.9 | 17.6 |
| 23 | B/32 | Experts Choice | 32 | 392 | – | 0.5 | 49.0 | 64.3 | 13.7 | 18.0 |
| 24 | B/32 | Experts Choice | 32 | 49 | – | 0.5 | 47.8 | 62.4 | 14.3 | 18.2 |
| 25 | S/16 | Experts Choice | 128 | 196 | – | 0.5 | 50.2 | 66.7 | 15.8 | 18.5 |
| 26 | S/16 | Experts Choice | 32 | 196 | – | 1.0 | 50.5 | 67.4 | 17.5 | 18.8 |
| 27 | S/16 | Experts Choice | 128 | 1568 | – | 0.5 | 51.4 | 67.7 | 16.8 | 19.7 |
| 28 | B/32 | Experts Choice | 32 | 392 | – | 1.0 | 49.9 | 66.0 | 16.5 | 19.7 |
| 29 | B/32 | Experts Choice | 64 | 392 | – | 1.0 | 49.9 | 65.5 | 16.5 | 19.8 |
| 30 | B/32 | Tokens Choice | 32 | 392 | 1 | 1.0 | 49.7 | 66.2 | 16.5 | 20.0 |
| 31 | B/32 | Tokens Choice | 64 | 392 | 1 | 1.0 | 49.8 | 65.6 | 16.5 | 20.2 |
| 32 | B/32 | Soft MoE | 64 | 49 | – | – | 51.8 | 70.7 | 17.8 | 20.3 |
| 33 | B/32 | Experts Choice | 64 | 49 | – | 1.0 | 48.6 | 64.0 | 17.7 | 20.3 |
| 34 | B/32 | Experts Choice | 32 | 49 | – | 1.0 | 48.4 | 63.8 | 17.7 | 20.5 |
| 35 | S/16 | Experts Choice | 32 | 1568 | – | 1.0 | 51.3 | 68.7 | 21.5 | 21.5 |
| 36 | S/16 | Soft MoE | 256 | 196 | – | – | 53.8 | 73.7 | 19.9 | 22.1 |
| 37 | S/16 | Tokens Choice | 32 | 1568 | 1 | 1.0 | 51.2 | 68.9 | 21.5 | 23.2 |
| 38 | S/16 | Experts Choice | 256 | 196 | – | 1.0 | 50.7 | 67.7 | 19.8 | 23.3 |
| 39 | S/16 | Experts Choice | 32 | 196 | – | 2.0 | 51.0 | 68.3 | 23.1 | 23.5 |
| 40 | B/32 | Tokens Choice | 32 | 392 | 2 | 1.0 | 50.2 | 67.4 | 22.0 | 23.6 |
| 41 | B/32 | Tokens Choice | 64 | 392 | 2 | 1.0 | 50.8 | 68.0 | 22.1 | 23.8 |
| 42 | S/16 | Tokens Choice | 64 | 1568 | 1 | 1.0 | 51.5 | 69.1 | 21.3 | 24.9 |
| 43 | S/16 | Experts Choice | 256 | 1568 | – | 1.0 | 52.3 | 69.7 | 21.7 | 25.5 |
| 44 | S/16 | Experts Choice | 32 | 1568 | – | 2.0 | 52.4 | 70.3 | 31.0 | 27.8 |
| 45 | S/16 | Tokens Choice | 32 | 1568 | 2 | 1.0 | 52.1 | 70.3 | 31.0 | 30.0 |
| 46 | B/32 | Tokens Choice | 64 | 392 | 2 | 2.0 | 51.2 | 70.0 | 33.2 | 30.4 |
| 47 | B/32 | Tokens Choice | 32 | 392 | 2 | 2.0 | 51.0 | 69.5 | 33.6 | 31.1 |
| 48 | S/16 | Tokens Choice | 64 | 1568 | 2 | 1.0 | 52.3 | 70.4 | 31.1 | 32.0 |
| 49 | S/16 | Tokens Choice | 32 | 1568 | 2 | 2.0 | 52.8 | 71.4 | 50.0 | 42.5 |
| 50 | S/16 | Tokens Choice | 64 | 1568 | 2 | 2.0 | 52.9 | 71.4 | 50.1 | 45.1 |

Table 9: Model runs from Section 3.3 (shown in Pareto plot) trained for 300k steps on JFT with inverse square root decay and 50k steps cooldown. We trained dense and MoE (Soft MoE, Tokens Choice, Experts Choice) models with sizes S/32, S/16, S/8, B/32, B/16, L/32, L/16 and H/14. Sorted by increasing training TPUv3 days.

| Ref | Model | Routing | Experts | Group Size | K | C | JFT P@1 | IN/10shot | Train exaFLOP | Train Days |
|-----|-------|---------|---------|-----------|---|---|---------|-----------|---------------|------------|
| 51 | B/16 | Dense | – | – | – | – | 52.0 | 71.8 | 64.8 | 45.2 |
| 52 | B/16 | Soft MoE | 128 | 196 | – | – | 55.3 | 77.0 | 59.0 | 46.8 |
| 53 | B/16 | Experts Choice | 128 | 1568 | – | 0.5 | 53.7 | 73.0 | 59.0 | 48.2 |
| 54 | B/16 | Experts Choice | 32 | 196 | – | 1.0 | 53.3 | 73.0 | 65.6 | 51.0 |
| 55 | B/16 | Experts Choice | 128 | 196 | – | 0.5 | 52.5 | 72.2 | 58.8 | 52.6 |
| 56 | L/32 | Dense | – | – | – | – | 51.3 | 70.9 | 55.9 | 54.9 |
| 57 | L/32 | Experts Choice | 32 | 392 | – | 0.5 | 52.3 | 71.2 | 47.4 | 55.2 |
| 58 | L/32 | Experts Choice | 32 | 49 | – | 0.5 | 51.1 | 70.6 | 49.8 | 55.7 |
| 59 | L/32 | Soft MoE | 32 | 49 | – | – | 53.5 | 75.0 | 49.8 | 56.0 |
| 60 | B/16 | Experts Choice | 32 | 1568 | – | 1.0 | 54.2 | 74.5 | 73.6 | 56.2 |
| 61 | B/16 | Tokens Choice | 32 | 1568 | 1 | 1.0 | 53.7 | 74.4 | 73.6 | 57.8 |
| 62 | B/16 | Experts Choice | 256 | 196 | – | 1.0 | 52.7 | 72.7 | 73.4 | 58.1 |
| 63 | B/16 | Soft MoE | 256 | 196 | – | – | 55.8 | 78.0 | 73.7 | 58.2 |
| 64 | B/16 | Tokens Choice | 64 | 1568 | 1 | 1.0 | 54.0 | 74.8 | 73.2 | 58.7 |
| 65 | L/32 | Experts Choice | 64 | 392 | – | 1.0 | 52.7 | 72.1 | 56.9 | 60.4 |
| 66 | B/16 | Experts Choice | 256 | 1568 | – | 1.0 | 53.9 | 73.5 | 73.8 | 60.5 |
| 67 | L/32 | Experts Choice | 32 | 392 | – | 1.0 | 52.7 | 71.7 | 56.8 | 60.6 |
| 68 | L/32 | Tokens Choice | 64 | 392 | 1 | 1.0 | 51.9 | 71.4 | 56.9 | 61.0 |
| 69 | L/32 | Tokens Choice | 32 | 392 | 1 | 1.0 | 52.3 | 71.7 | 57.1 | 61.6 |
| 70 | L/32 | Experts Choice | 64 | 49 | – | 1.0 | 51.1 | 70.7 | 61.6 | 62.6 |
| 71 | L/32 | Soft MoE | 64 | 49 | – | – | 54.0 | 75.2 | 61.7 | 62.8 |
| 72 | L/32 | Experts Choice | 32 | 49 | – | 1.0 | 51.4 | 70.3 | 61.5 | 63.2 |
| 73 | B/16 | Experts Choice | 32 | 196 | – | 2.0 | 53.1 | 73.9 | 86.8 | 64.2 |
| 74 | L/32 | Tokens Choice | 32 | 392 | 2 | 1.0 | 51.5 | 70.7 | 76.0 | 72.2 |
| 75 | B/16 | Experts Choice | 32 | 1568 | – | 2.0 | 54.6 | 75.6 | 102.9 | 72.5 |
| 76 | L/32 | Tokens Choice | 64 | 392 | 2 | 1.0 | 52.0 | 71.8 | 76.0 | 72.5 |
| 77 | B/16 | Tokens Choice | 32 | 1568 | 2 | 1.0 | 53.9 | 74.7 | 102.9 | 74.7 |
| 78 | B/16 | Soft MoE | 512 | 196 | – | – | 56.1 | 78.5 | 103.1 | 76.5 |
| 79 | B/16 | Tokens Choice | 64 | 1568 | 2 | 1.0 | 54.3 | 74.8 | 103.0 | 76.5 |
| 80 | S/8 | Dense | – | – | – | – | 49.9 | 66.7 | 82.7 | 77.7 |
| 81 | S/8 | Soft MoE | 512 | 784 | – | – | 56.1 | 78.0 | 85.6 | 88.5 |
| 82 | S/8 | Experts Choice | 32 | 784 | – | 1.0 | 52.9 | 72.6 | 91.3 | 93.0 |
| 83 | L/32 | Tokens Choice | 64 | 392 | 2 | 2.0 | 52.9 | 72.9 | 114.3 | 93.2 |
| 84 | L/32 | Tokens Choice | 32 | 392 | 2 | 2.0 | 52.5 | 72.5 | 115.7 | 95.8 |
| 85 | L/16 | Dense | – | – | – | – | 54.8 | 77.8 | 226.9 | 100.9 |
| 86 | L/16 | Experts Choice | 128 | 196 | – | 0.5 | 54.0 | 76.7 | 204.6 | 104.9 |
| 87 | L/16 | Soft MoE | 128 | 196 | – | – | 57.2 | 80.3 | 205.0 | 106.0 |
| 88 | B/16 | Tokens Choice | 32 | 1568 | 2 | 2.0 | 54.4 | 75.7 | 161.4 | 108.4 |
| 89 | B/16 | Tokens Choice | 64 | 1568 | 2 | 2.0 | 54.8 | 76.0 | 161.5 | 110.5 |
| 90 | L/16 | Experts Choice | 32 | 196 | – | 1.0 | 55.1 | 77.5 | 228.6 | 113.6 |
| 91 | L/16 | Tokens Choice | 32 | 1568 | 1 | 1.0 | 55.9 | 78.5 | 250.4 | 125.1 |
| 92 | L/16 | Tokens Choice | 64 | 1568 | 1 | 1.0 | 56.2 | 78.6 | 248.8 | 125.7 |
| 93 | S/8 | Experts Choice | 32 | 6272 | – | 1.0 | 53.6 | 73.4 | 160.6 | 126.6 |
| 94 | S/8 | Experts Choice | 512 | 784 | – | 1.0 | 53.4 | 72.4 | 104.1 | 129.0 |
| 95 | L/16 | Soft MoE | 256 | 196 | – | – | 57.4 | 80.2 | 256.0 | 129.6 |
| 96 | S/8 | Tokens Choice | 32 | 6272 | 1 | 1.0 | 53.8 | 73.7 | 162.5 | 129.8 |
| 97 | L/16 | Experts Choice | 256 | 196 | – | 1.0 | 54.1 | 76.7 | 255.2 | 130.1 |
| 98 | L/16 | Experts Choice | 32 | 196 | – | 2.0 | 55.2 | 77.8 | 301.0 | 140.3 |
| 99 | S/8 | Experts Choice | 512 | 6272 | – | 1.0 | 54.8 | 74.6 | 149.3 | 161.9 |
| 100 | S/8 | Tokens Choice | 32 | 6272 | 2 | 1.0 | 54.2 | 74.6 | 243.4 | 166.6 |
| 101 | H/14 | Soft MoE | 128 | 256 | – | – | 58.0 | 81.6 | 599.2 | 170.5 |
| 102 | H/14 | Dense | – | – | – | – | 56.5 | 80.1 | 680.5 | 196.2 |

Table 9: Model runs from Section 3.3 (shown in Pareto plot) trained for 300k steps on JFT with inverse square root decay and 50k steps cooldown. We trained dense and MoE (Soft MoE, Tokens Choice, Experts Choice) models with sizes S/32, S/16, S/8, B/32, B/16, L/32, L/16 and H/14. Sorted by increasing training TPUv3 days.

| Ref | Model | Routing | Experts | Group Size | K | C | JFT P@1 | IN/10shot | Train exaFLOP | Train Days |
|-----|-------|---------|---------|------------|---|------|---------|-----------|---------------|------------|
| 103 | H/14 | Experts Choice | 64 | 2048 | – | 1.25 | 57.3 | 80.4 | 855.9 | 210.9 |
| 104 | L/16 | Tokens Choice | 32 | 1568 | 2 | 2.0 | 53.5 | 74.6 | 534.5 | 218.5 |
| 105 | L/16 | Tokens Choice | 64 | 1568 | 2 | 2.0 | 53.3 | 73.3 | 535.1 | 226.9 |
| 106 | H/14 | Tokens Choice | 64 | 2048 | 1 | 1.25 | 56.7 | 79.8 | 857.0 | 230.7 |
| 107 | S/8 | Tokens Choice | 32 | 6272 | 2 | 2.0 | 54.1 | 74.8 | 424.4 | 255.4 |