# OpenReview forum: "From Sparse to Soft Mixtures of Experts"
_ICLR.cc/2024/Conference — ICLR 2024 spotlight_

### Official Review · Reviewer_pKE3 · 2023-10-29

**Soundness:** 3 good
**Presentation:** 2 fair
**Contribution:** 3 good
**Rating:** 8
**Confidence:** 3

**Summary:**

This work focuses on a number of issues in vanilla Sparse MoEs, and proposes Soft MoE. Instead of hard assignment of tokens, Soft MoE “softs” the tokens by weighted averaging them before assigned to experts. Experiments with ViT models and CV tasks demonstrate that Soft MoE outperforms both Sparse MoEs and dense models.

**Strengths:**

This work is novel and sound. The soft assignment by learnable weighted combinations helps tackle many limitations of vanilla Sparse MoEs. Extensive empirical results show that Soft MoE outperforms vanilla Sparse MoEs and dense models in terms of model ability (precisions/accuracies/...) and improves the training and inference speed/throughput compared with dense models.

**Weaknesses:**

The proposed technique cannot be integrated with main-stream foundation models (e.g., auto-regressive LLMs), which may limit the applications.

**Questions:**

When assessing the training and inference speed, the authors chose dense ViT as the baseline. I would be better to include the results for Sparse MoEs additionally.

The experiments focus on CV tasks (in particular, ViT-based MoEs). It makes me wonder how Soft MoE would be on NLP tasks. Although in Section 6 the authors have discussed that Soft MoE is difficult to use in auto-regressive models, attempts on BERT- or T5-based MoEs should be suggested.

---

> ### Author Response · Authors · 2023-11-23
>
> Please, refer to our comment addressed to all reviewers: https://openreview.net/forum?id=jxpsAj7ltE&noteId=PeYDtrMiSS

---

### Official Review · Reviewer_wHYN · 2023-10-31

**Soundness:** 4 excellent
**Presentation:** 4 excellent
**Contribution:** 3 good
**Rating:** 8
**Confidence:** 4

**Summary:**

Instead of sending each expert a bag of vectors from the input the vectors entering each expert are composed of a weighted sum of all input vectors. My understanding of the implementation was assisted by [Phil Wang's replication][lucidrains] in PyTorch, which I have shortened below:

```
def forward(self, x):
        """
        einstein notation
        b - batch
        n - sequence length
        e - number of experts
        s - number of slots per expert
        d - feature dimension
        """
        slot_embeds = self.slot_embeds

        logits = einsum('b n d, e s d -> b n e s', x, slot_embeds)

        # get dispatch and combine weights (softmax across right dimensions)

        dispatch_weights = logits.softmax(dim = 1)

        combine_weights = rearrange(logits, 'b n e s -> b n (e s)')
        combine_weights = combine_weights.softmax(dim = -1)

        # derive slots by weighted average of input tokens using the dispatch weights from above

        slots = einsum('b n d, b n e s -> b e s d', x, dispatch_weights)

        # route the slots per expert to each expert

        out = self.experts(slots) # vectors are mapped across dim e to each expert

        # combine back out

        out = rearrange(out, ' b e s d -> b (e s) d')
        out = einsum('b s d, b n s -> b n d', out, combine_weights)
```

The contributions of this paper are mainly the experimental results, the experiments aim to demonstrate that a soft-MoE layer allows faster inference and shorter training time on some large scale training benchmarks. Specifically:

- Soft MoE L/16 vs ViT H/14:
    - Half the training time
    - 2x faster inference
- Soft MoE B/16 vs ViT/14:
    - 5.7x faster at inference time

These results are demonstrated comprehensively with ablation experiments investigating the different assumptions, such as whether the weighted sum is necessary, whether the weighted sum can be uniform and whether the weighted sum at the output is necessary; finding that the decisions all improve performance.

The results are demonstrated on expensive large scale benchmarks, including training many ViT style models on JFT-4B and further experiments testing these models on a contrastive benchmark using WebLI. The results presented support the contributions of the paper, demonstrating improved performance across the experiments compared to other MoE methods.

The authors also note that the proposed method would violate the causal assumption if used in an autoregressive model, meaning it can't be used as defined above for autoregressive language modelling.

[lucidrains]: https://github.com/lucidrains/soft-moe-pytorch/blob/main/soft_moe_pytorch/soft_moe.py

**Strengths:**

The paper makes a clear contribution of an architectural improvement that requires extensive experimental justification to prove. It then provides the experimental results to back up this assertion on expensive benchmarks. The results are presented clearly and it is easy for the reader to find relevant information.

**Weaknesses:**

Minor presentation issue:

- Figure 2 is confusing, the relationship between dispatch and combine weights to the tokens is illustrated with two downward arrows, but they don't really mean anything so it doesn't help the reader
- Better signposting about all the different results that may be found in the experiments section as it is the most valuable part of the paper. For example, in the introduction some of the inference time and training time benefits are mentioned but not where these results are found.

It would be useful to see the complexity results described in Section 2.3 verified by experiment, perhaps looking at throughput in practice. I think Figure 6 implies something to do with this but the relationship is not entirely obvious.

**Questions:**

The authors mention that autoregessive modelling is an area for future work, but it's not totally clear to me how to achieve that with this type of model. The Token mixing weights would need to have a causal mask similar to the attention causal mask, but that seems like it might affect performance or throughput. Do they have any ideas how to approach this issue?

---

> ### Author Response · Authors · 2023-11-23
>
> Please, refer to our comment addressed to all reviewers: https://openreview.net/forum?id=jxpsAj7ltE&noteId=PeYDtrMiSS

---

### Official Review · Reviewer_fyPo · 2023-11-01

**Soundness:** 4 excellent
**Presentation:** 4 excellent
**Contribution:** 4 excellent
**Rating:** 8
**Confidence:** 4

**Summary:**

This paper introduces a novel soft MOE method that keeps the benefits of sparse MOE (processing a subset of tokens for lower cost), and also has full differentiability and better performance. The proposed method let the MOE experts to process a number of slots, where each slot is a combination of all tokens. The idea of processing a number of slots is how the inference cost is controlled. It outperforms dense (non MOE) networks and also typical sparse MOE networks: with similar inference cost, the accuracy metric is much better.

**Strengths:**

Originality
The idea of this soft MOE is new. I have not encountered previous work that uses this idea. It combines the idea of sparse MOE with an attention-like mechanism to get the benefits of both. The major part that I like is that this combination is very practical in large scale model that requires model parallelism. The author also clearly contrasted its difference with other existing works and multi-head attention in Section 5.

Quality
The quality of the paper is high.
-	The author provides very good details of how the method is implemented and some details of implementation.
-	The experiments settings are clearly stated and the hyper-parameter searching is transparent and properly done for contrast models (dense models and other MOE models) as well.
-	The design choices (slots per token and expert numbers) are well studied and concluded.
-	The experiments conducted are at large scale datasets and on various tasks/settings, and the improvement of the proposed methods are convincingly large.

Clarity
The clarity of the paper is good. The demonstration of the method and experiment conclusions are clear and well supported by experiments.

Significance
This is very exciting work that can impact the community greatly. The proposed method shows convincing performance dominancy over other MOE methods or dense models on image classification task and contrastive learning tasks. It shows that with similar inference cost, it can achieve much better accuracy metric, even against strong baselines. It is also very practical that it can be used in modern large scale parallel learning platforms without too much extra optimization needed.

**Weaknesses:**

as mentioned already by the author, the main weakness of this method from my perspective is: each expert do not handle multiple tokens well (i.e. one expert one token is better). In practice, this may cause inefficient increase of number of parameters (memory). But the authors have already recognized it, and I think it doesn't hurt the significance of the existing contribution of this paper.

**Questions:**

I hope the authors can release as much details as possible of the implementation details. Looking forward to the adaptation into different fields.

---

> ### Author Response · Authors · 2023-11-23
>
> Please, refer to our comment addressed to all reviewers: https://openreview.net/forum?id=jxpsAj7ltE&noteId=PeYDtrMiSS

---

### Official Review · Reviewer_HCxh · 2023-11-08

**Soundness:** 3 good
**Presentation:** 4 excellent
**Contribution:** 4 excellent
**Rating:** 6
**Confidence:** 3

**Summary:**

The following paper proposes Soft Mixture-of-Experts (MoE), a fully-differentiable sparse transformers capable of passing different weighted combinations of all input tokens to each expert. Compared to other MoEs, experts in Soft MoE only process a subset of (combined) tokens, resulting in larger model capacity (and performance) with lower inference cost. Thanks to this, Soft MoE are able to alleviate training instability, token dropping, scalability issues, and ineffective finetuning process that is apparent in previous type of MoE. Experiments on visual recognition tasks in Section 3 and 4 show the superiority of Soft MoE over dense Vision Transformers (ViT) and popular MoEs not limited to Tokens Choice and Experts Choice. In addition to that, Soft MoE provably scales well; since while the Huge/14 version of it has $40\times$ more parameters than ViT Huge/14, it bring substantial improvement in terms of classification accuracy with only 2% gain in inference.

**Strengths:**

- Paper is well-written, easy to understand, and provided with source codes that model how Soft MoE is being implemented.
- Experiments are comprehensive and detailed with respect to its domain, with emphasis not only on performance, but also in terms of inference and wall-clock time.
- High potential to bring improvement when it is deployed towards various modalities such as Large-Language Models (LLMs).

**Weaknesses:**

Aside from the weaknesses mentioned in the paper, I would like to address concerns that is apparent in the paper:
- The experiments performed in Sections 3 and 4 seem to focus only on vision-related tasks. It would be great to be able to observe results on different modalities such as NLP-related tasks based on GLUE or SuperGLUE benchmark that is performed in [1].
- Unfortunately, the dataset that is being used for training is not publicly available; making it hard to be used for benchmarking with other papers.
- As mentioned in Section 6, the memory footprint will be huge when we leverage a large number of experts, and usage in auto-regressive decoders is problematic due to the preservation of past and future tokens during training.

**Questions:**

- Is it possible to see the result of MoE in GLUE or SuperGLUE benchmark as seen on [1]?

[1] Zhou et al. Mixture-of-experts with expert choice routing. Advances in Neural Information Processing Systems, 35:7103–7114, 2022. https://arxiv.org/abs/2202.09368

---

> ### Author Response · Authors · 2023-11-23
>
> Please, refer to our comment addressed to all reviewers: https://openreview.net/forum?id=jxpsAj7ltE&noteId=PeYDtrMiSS

---

### Author Response · Authors · 2023-11-23
**Response to all reviewers**

We are glad of the mostly positive comments and scores from the reviewers, and we are very thankful for their very useful comments to help to improve our submission.

- All reviewers commented on the lack of natural language processing results and/or the challenges of applying Soft MoEs in this domain.

Indeed, NLP and text auto-regressive tasks in particular, are a highly important application in current ML trends. We want to emphasize that we are actively working on this, but we decided to limit the scope of the paper to vision tasks for a few reasons:

1. By narrowing the scope of domains, we can do a more in-depth analysis of Soft MoEs in vision tasks (see all the detailed investigation in the appendix).
2. Limiting the scope of tasks also makes the presentation of the experiments, and the overall reading and understanding of the model clearer (in our opinion), and allows one to focus on the fundamental aspects of the proposed method (we are glad that most reviewers value the quality of the presentation very highly).
3. We want to present our current (very good) results to the ML community, and encourage it to explore and improve the proposed approach for NLP and many other modalities and tasks. Certainly, tens or hundreds of researchers can make faster progress than just a handful, we believe.


- Reviewer HCxh commented:

>  Unfortunately, the dataset that is being used for training is not publicly available; making it hard to be used for benchmarking with other papers.


We have used the rebuttal period to run additional experiments pre-training on the publicly available LAION-400M dataset (using SigLIP pre-training, as described in [1]). The results are consistent with those obtained on JFT. Other researchers will be able to replicate our results, since our code is open-sourced and we will release the configuration file to run the LAION pre-training.

Here we show different downstream results obtained from LAION-400M pre-trained models:
https://drive.google.com/file/d/16EN8jDbXxLC4-CGxWpBeP0EuU4dBIIva/view

As the plot shows, Soft MoE has better quality than both ViT and Experts Choice, when matching the number of steps (which also roughly matches training cost). We will include these (and more detailed) results obtained on this publicly available dataset in the final version of the paper, most likely in the appendix due to space limitations.

- Reviewer pKE3 said:

> When assessing the training and inference speed, the authors chose dense ViT as the baseline. It would be better to include the results for Sparse MoEs additionally.

We compare quality vs. comparing training and inference speed of Sparse MoEs and Dense in Figure 3 (main paper) and Table 9 (appendix J). For long training runs, we compare Soft MoE only with the already published results of ViT in [2], since these runs are very costly and we couldn’t afford to train all methods for millions of steps.

- Reviewer wHYN made some comments on the presentation.

Regarding Figure 2: we tried to imply that the dispatch and combine weights were used to map input tokens to slots (dispatch) and slots to output tokens (combine). We’ll try to find a better representation of this, or annotate the image for further clarification.

Regarding time complexity: indeed Figure 6 shows the fact that the time complexity of Sparse MoE is higher than Soft MoEs, due to the sorting operations: when the number of experts grows a lot, the sorting operations start to dominate the runtime and the throughput decreases a lot for Experts Choice and Tokens Choice. It’s a bit hard to visualize the time complexity in a single plot, since it depends on multiple factors such as the number of experts, the group size, and the model dimension. We’ll add additional plots in the appendix trying to visualize these dependencies.

We’ll also add additional references to where to find the supporting results when claims are made.

- Reviewer fyPo commented:

> I hope the authors can release as much details as possible of the implementation details. Looking forward to the adaptation into different fields.

We have already open-sourced our implementation, including details about the training hyperparameters used in this paper. We have excluded the link to the code in the draft to avoid breaking double-blind review process, but this will be part of the final version. We will also include the configuration files to replicate the LAION experiments mentioned above.

We hope that we have addressed your questions and concerns. Let us know if any further clarification is needed.


[1] Sigmoid Loss for Language Image Pre-Training, https://arxiv.org/abs/2303.15343

[2] Scaling Vision Transformers, https://arxiv.org/abs/2106.04560

---

### Meta-Review · Area_Chair_iifn · 2023-12-06

**Metareview:**

The paper introduces Soft Mixture-of-Experts (MoE), a novel approach that combines the benefits of sparse MoEs with full differentiability, addressing issues such as training instability, token dropping, scalability, and finetuning challenges seen in previous MoE models. Soft MoE allows experts to process a subset of tokens, reducing inference costs while maintaining model capacity and performance. Experimental results show that Soft MoE significantly outperforms dense networks and conventional sparse MoE models in terms of classification accuracy while offering faster inference and shorter training times, particularly in comparison to ViT models.

The reviewers were concerned lack of natural language processing applications, lack of results on public benchmark datasets, and choice of baseline.

Despite the concerns, all the reviewers are positive.
Most of the reviewers' concerns and questions have been well-addressed by the authors through the rebuttal.

The highlight of this work is that the authors demonstrate a simple design choice can help effectively tackle many limitations of vanilla Sparse MoEs.

**Justification For Why Not Higher Score:**

While the idea is simple and effective (which is good), the approach can be applied to limited architectures, e.g., ViT, and applications, e.g., non-autoregressive tasks (NLP). This may limit the interest of the community.

**Justification For Why Not Lower Score:**

This work got positive responses from the reviewers.
The authors demonstrate the proposed method helps effectively tackle many limitations of vanilla Sparse MoEs.

---

### Decision · Program_Chairs · 2024-01-16

Accept (spotlight)